# Rapid and quantitative functional interrogation of human enhancer variant activity in live mice

Ethan W. Hollingsworth [1,2], Taryn A. Liu [1], Joshua A. Alcantara [1], Cindy X. Chen[1], Sandra H. Jacinto[1] & Evgeny Z. Kvon [1]✉

Functional analysis of non-coding variants associated with congenital disorders remains challenging due to the lack of efficient in vivo models. Here we introduce dual-enSERT, a robust Cas9-based two-color fluorescent reporter system which enables rapid, quantitative comparison of enhancer allele activities in live mice in less than two weeks. We use this technology to examine and measure the gain- and loss-of-function effects of enhancer variants previously linked to limb polydactyly, autism spectrum disorder, and craniofacial malformation. By combining dual-enSERT with single-cell transcriptomics, we characterise gene expression in cells where the enhancer is normally and ectopically active, revealing candidate pathways that may lead to enhancer misregulation. Finally, we demonstrate the widespread utility of dual-enSERT by testing the effects of fifteen previously uncharacterised rare and common non-coding variants linked to neurodevelopmental disorders. In doing so we identify variants that reproducibly alter the in vivo activity of *OTX2* and *MIR9-2* brain enhancers, implicating them in autism. Dual-enSERT thus allows researchers to go from identifying candidate enhancer variants to analysis of comparative enhancer activity in live embryos in under two weeks.

The success of the large-scale genome-wide association (GWAS) and whole-genome sequencing (WGS) studies has shifted the bottleneck of human genetics from identifying sources of genetic variation to mechanistically understanding how such variation contributes to human disease[1–3]. Nearly 90% of disease risk-associated variation resides in non-protein coding regions of the human genome[4–8]. A large fraction of this variation consists of single nucleotide polymorphisms and rare variants that are hypothesised to affect transcriptional enhancers, short non-coding DNA segments that regulate cell-type-specific gene expression[9–13]. Each of these thousands of enhancer variants thus represents a potential entry point for understanding human disease[1–3]. However, the physiological effects of the vast majority of these associations remain unknown. Bridging this gap – from non-coding variants to biological mechanisms – is currently hindered by a lack of suitable in vivo

technologies for assessing if and how each human enhancer variant alters gene expression.

A major challenge is that the effects of enhancer variants on gene expression are highly cell-type-specific. For example, a typical gain-of-function enhancer variant can result in ectopic gene expression and cause pathogenic effects in cells where the enhancer is normally inactive[14–21]. Likewise, loss-of-function enhancer variants often result in loss of enhancer activity in one cell type, while in other cell types, its activity is unaffected[22–26]. These cell-type-specific effects of enhancer variants are difficult to capture with high-throughput methods such as massively parallel reporter assays (MPRAs) and CRISPR inhibitor/activator screens, both of which are primarily performed in vitro[27,28] or in one tissue[29–34]. Transgenic enhancer-reporter assays in mice enable visualisation of enhancer activity in the whole animal and are a gold standard for functionally testing when and where a human enhancer is

[1]Department of Developmental and Cell Biology, University of California, Irvine, CA, USA. [2]Medical Scientist Training Program, University of California, Irvine School of Medicine, Irvine, CA, USA. ✉e-mail: ekvon@uci.edu

active in vivo[35–37]. However, current transgenic mouse reporter assays only allow the assessment of a single enhancer per animal, precluding the direct comparison of multiple enhancer alleles. Thus, comparing activities of reference and disease-linked variant alleles requires the generation of a large number of independent transgenic mice to mitigate variation caused by mosaicism and position effects[17,20,38].

Here, we introduce dual-enSERT (dual-fluorescent enhancer inSERTion), a Cas9-based site-specific dual-fluorescent reporter system that enables simultaneous, quantitative visualisation of two human enhancer allelic activities in the same transgenic animal, thus overcoming the limitations of standard mouse reporter assays. In dual-enSERT-1, transgenes containing enhancer variants driving *eGFP* or *mCherry* are placed on different alleles of the same safe-harbour location in the mouse genome. In dual-enSERT-2, both enhancer-reporters are placed on the same transgene separated by a synthetic insulator. Using dual-enSERT-2, we were able to visualise and compare two enhancer allelic activities in live F0 mice as soon as eleven days after zygote microinjection. We first applied dual-enSERT to previously characterised pathogenic enhancer variants linked to autism spectrum disorder, limb defects, and craniofacial malformation, confirming the reported loss- and gain-of-function effects on in vivo enhancer activity. To demonstrate the utility of dual-enSERT for screening untested non-coding variants, we interrogated a panel of fifteen previously uncharacterised non-coding variants from patients with neurodevelopmental disorders and identified variants that reproducibly alter enhancer activity in vivo. Beyond the quantitative visualisation of enhancer activity, coupling dual-enSERT with single-cell transcriptomics enables the characterisation of gene expression and pathogenic enhancer activity at cellular resolution. Our dual-enSERT system is thus poised to accelerate enhancer-variant-to-function studies across many congenital disorders.

## Results

### Direct comparison of reference and variant enhancer allele activities in vivo with dual-enSERT-1

Classical mouse enhancer-reporter assays are based on random integration of the transgene into the genome[35–37,39,40]. Although conventional mouse transgenesis is the current gold standard for visualisation of enhancer activity in vivo, it suffers from variation due to position effects and requires the generation of a large number of transgenic animals to assess enhancer activity reproducibly[17,41,42]. Enhancer-reporter assays based on the integration of a transgene into a safe-harbour location of the genome overcome the problem of position effects[20,43]. However, due to mosaicism and inter-embryo variability, these site-specific assays also require analysis of a large number of transgenic animals, especially for detecting the subtle effects of enhancer mutations[20,38,44]. To overcome these limitations, we developed dual-enSERT-1, a transgenic approach based on highly efficient Cas9-mediated integration of enhancers driving fluorescent reporters into the H11 safe-harbour integration site[45]. One enhancer allele is placed upstream of an *eGFP* reporter and the second enhancer allele is placed upstream of a *mCherry* reporter followed by Cas9-mediated integration of each transgene into the H11 locus (Fig. 1a). With dual-enSERT-1, we achieved an average transgenic targeting efficiency of 57% across all single-reporter constructs tested in this study, and all insertions were germline transmissible (Fig. 1b).

We first assessed whether we could detect and quantify the effects of non-coding variants on enhancer activity with dual-enSERT-1 by testing a previously characterised pathogenic allele of the ZRS (zone of polarising activity (ZPA) Regulatory Sequence, also known as MFCS1) enhancer of *Sonic hedgehog (Shh)*. Single nucleotide variants in the ZRS cause congenital limb malformations, most typically preaxial polydactyly, in humans, cats, chickens, and mice[20,46,47]. ZRS variants

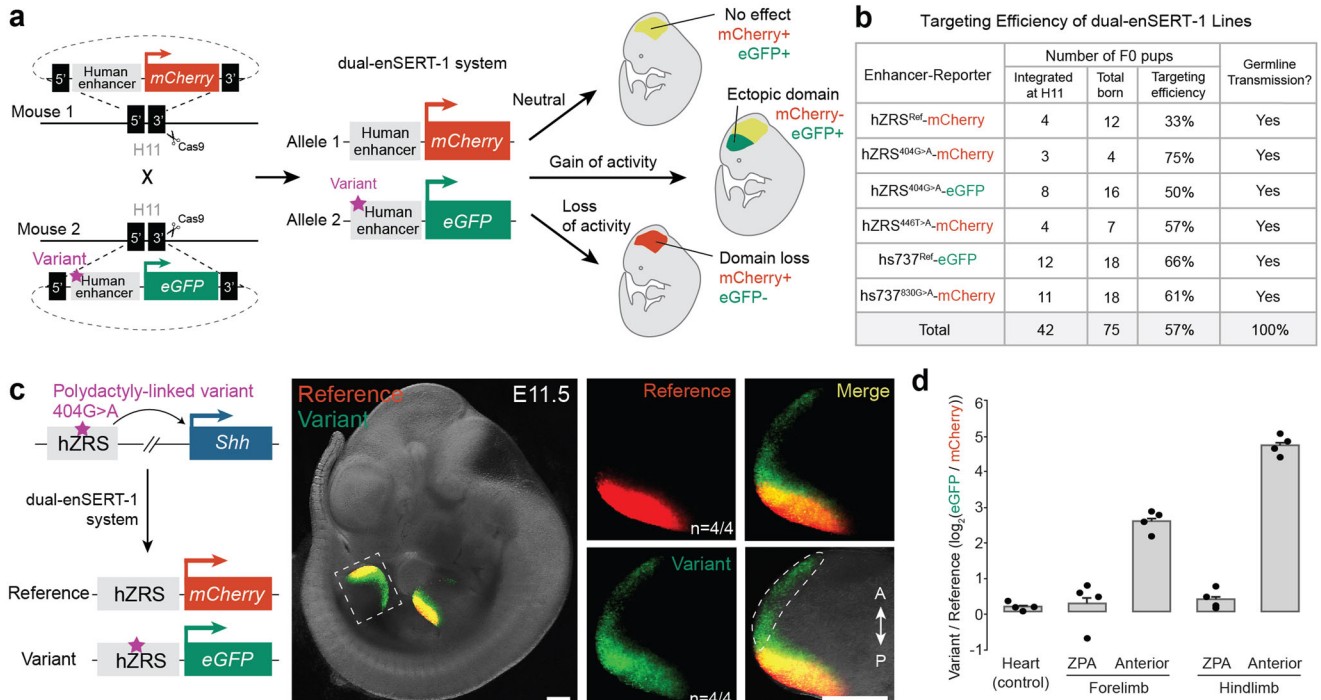

**Fig. 1 | Simultaneous comparison of human reference and variant enhancer activities in developing mouse embryos. a** Schematic overview of the dual-enSERT-1 strategy and its potential readouts for different types of enhancer variants. **b** Transgenic targeting efficiencies of all dual-enSERT-1 constructs generated in this study. **c** Representative images of transgenic hZRS^ref^-*mCherry*/hZRS^404G>A^-*eGFP* embryos at E11.5. A close-up of the hindlimb with separate and merged channels and an outline depicting the gain of enhancer activity (green channel) in

the anterior domain are shown. A, anterior; P, posterior. **d** Plots quantifying fold-change (log₂) difference in normalised reporter fluorescence between variant and reference hZRS alleles. Two-sided paired *t* tests vs. Heart: Forelimb ZPA, *P* = ns; Forelimb Anterior, *P* = 0.00072; Hindlimb ZPA, *P* = ns; Hindlimb Anterior, *P* = 9.48E-08. Data represented as mean ± SEM. Data points represent independent biological replicates (*n* = 4 embryos). Scale bars, 500 μm. Source data are provided as a Source Data file.

implicated in preaxial polydactyly cause ectopic *Shh* expression in the anterior portion of the developing limb bud, leading to erroneous digit outgrowth similar to human patients[20,47]. We created two stable transgenic mouse lines, one with the human reference ZRS allele driving *mCherry* (hZRS^ref-*mCherry)* and a second line with a previously characterised pathogenic hZRS allele containing the polydactyly-linked 404G>A variant driving *eGFP* (hZRS^404G>A-*eGFP*) (Fig. 1c). To visualise reference and variant hZRS enhancer activities simultaneously, we crossed these mouse lines to generate two-colour dual-enSERT-1 embryos. Both lines had two copies of respective transgene integrated at the H11 locus enabling direct and quantitative comparison between enhancer alleles in two-colour dual-enSERT-1 embryos (Supplementary Fig. 1a, b and "Methods"). In these hZRS^ref-*mCherry*/hZRS^404G>A-*eGFP* transgenic embryos, mCherry fluorescence was detected in the ZPA of fore- and hindlimb buds, matching the location of normal *Shh* expression ($n = 4/4$ embryos)[48]. We also detected weaker mCherry fluorescence elsewhere in the embryo, including the heart, consistent with the weak endogenous activity of the *Hsp68* promoter[20]. EGFP expression pattern driven by the hZRS^404G>A variant allele was indistinguishable from the mCherry expression pattern except for the anterior limb region. eGFP fluorescence extended into the anterior domain of the limb bud in all examined embryos, mimicking ectopic *Shh* misexpression in mice with polydactyly ($n = 4/4$; Fig. 1c, d).

To quantify the effect of 404G>A variant on hZRS enhancer activity, we compared eGFP and mCherry fluorescent intensities within the same embryo. We detected similar levels of eGFP and mCherry fluorescence in the heart (1.1-fold difference, eGFP vs. mCherry, $P = ns$), indicating that differences in eGFP and mCherry maturation times and half-lives have a negligible effect on our measurements. Nevertheless, to exclude even small confounding effects caused by differences in the choice of the reporter, we used promoter-driven heart fluorescence as an endogenous control for all future comparisons ("Methods"). Both alleles drove similar levels of reporter expression in the ZPA (Forelimb, $P = ns$; Hindlimb, $P = ns$). We detected a 6.5-fold stronger reporter expression in the anterior forelimb bud ($P = 0.00072$) and a 31-fold stronger expression in the anterior hindlimb bud ($P = 9.84E-08$). These results indicate that dual-enSERT-1 can robustly detect and quantify changes in limb enhancer activity caused by pathogenic hZRS variants.

We next asked whether dual-enSERT-1 could be used to study human non-coding variants linked to other congenital disorders. We focused on the hs737 enhancer of *EBF3*, a region where several independent rare variants have been identified in patients with autism spectrum disorder and intellectual disability[17,49]. We generated a transgenic mouse line in which the human reference hs737 allele drives *eGFP* (hs737^ref-e*GFP*) and a second line in which an 830G>A variant allele, identified in a patient with autism, drives *mCherry* (hs737^830G>A-*mCherry)* (Supplementary Fig. 1c). We then bred these mouse lines each containing single-copy integrated transgenes and examined reporter gene expression in E11.5 embryos (Supplementary Fig. 1a, b). Live imaging revealed comparable levels of eGFP and mCherry fluorescence in the midbrain, hindbrain, and neural tube (Midbrain, $P = ns$; Hindbrain, $P = ns$; Neural Tube, $P = ns$; Supplementary Fig. 1c, d). However, mCherry expression driven by the hs737^830G>A variant allele also extended into the forebrain in all examined hs737^ref-e*GFP*/hs737^830G>A-*mCherry* embryos (9.4-fold difference in Forebrain, $P = 0.0043$; Supplementary Fig. 1c, d). These results are consistent with previous observations of ectopic forebrain activity using non-quantitative LacZ-based transgenic assays[17].

## Comparative functional assessment of independent enhancer variants

We next tested the ability of dual-enSERT-1 to simultaneously visualise and compare the effects of independent enhancer variants in live mice. Human genetics studies often identify disease-linked hotspots in which multiple rare variants affect the same enhancer[11,20,50–53]. For example, 22 different rare human point mutations in the hZRS enhancer have been identified in patients with polydactyly[20,47]. Despite extensive work on this enhancer, it is unknown if these independent mutations result in ectopic gene expression in the same or different cell populations of the limb bud.

As a positive control, we first created a mouse line in which hZRS^404G>A allele drives *mCherry* and crossed this line to the mouse line in which the same hZRS^404G>A allele drives *eGFP*. We collected the resulting two-colour hZRS^404G>A-*mCherry*/hZRS^404G>A-*eGFP* transgenic embryos with the expectation that most fluorescent cells will be double positive eGFP + /mCherry + cells (Fig. 2a and Supplementary Fig. 2a). Indeed, at E11.5, the overlap in ectopic GFP and mCherry activity in the anterior limb bud mesenchyme was visually indistinguishable (Fig. 2a and Supplementary Fig. 2a). To quantify this overlap at cellular resolution, we used fluorescence-activated cell sorting (FACS) to isolate double-positive cells from anterior limb bud mesenchyme (Fig. 2c). As a negative control, we used transgenic mice in which the ZRS^404G>A variant allele was driving *eGFP* and the ZRS reference allele was driving *mCherry* (hZRS^ref-*mCherry*/hZRS^404G>A-*eGFP*) with the expectation that only eGFP+ cells should be present in the anterior limb bud cell population (Fig. 1c). Indeed, 92% of fluorescent cells sorted from anterior limb buds of hZRS^ref-*mCherry*/hZRS^404G>A-*eGFP* embryos were eGFP+/mCherry- and only 3% were eGFP+/mCherry+, confirming allele-specific ectopic expression of the ZRS^404G>A allele (Fig. 2c, d and Supplementary Fig. 3a, b). We next examined hZRS^404G>A-*mCherry*/hZRS^404G>A-*eGFP* transgenic embryos carrying the ZRS^404G>A variant allele driving both colours. Only 56% (in forelimbs) to 53% (in hindlimbs) of fluorescent cells in anterior limb buds were eGFP + /mCherry + (Fig. 2d and Supplementary Fig. 3c).

To test if this variability could be caused by differences between fluorophores, we performed *mCherry* and *eGFP* mRNA quantification in different populations of anterior limb bud cells. We dissected the anterior domains of hindlimbs from hZRS^404G>A-*mCherry*/hZRS^404G>A-*eGFP* embryos and sorted them into four cell populations: mCherry + /eGFP +, mCherry + /eGFP-, mCherry-/eGFP +, and mCherry-/eGFP-. We then performed qPCR to quantify *mCherry* and *eGFP* mRNA levels in each of these cell populations (Supplementary Fig. 3d). Both eGFP-positive cell populations (mCherry + /eGFP + and mCherry-/eGFP +) expressed more than 8-fold higher levels of *eGFP* than eGFP-negative cell populations (mCherry + /eGFP + vs mCherry-/eGFP-, $P = 0.008$; mCherry + /eGFP + vs mCherry + /eGFP-, $P = 0.009$) (Supplementary Fig. 3d). This indicates that eGFP fluorescence accurately reflects *eGFP* expression. Both mCherry-positive cell populations displayed comparable levels of *eGFP* and *mCherry* transcripts ($P = NS$). However, similar levels of *mCherry* transcripts were also observed in mCherry-negative cells ($P = NS$) (Supplementary Fig. 3d). The incomplete fluorescent overlap on a single-cell level is likely due to post-transcriptional differences between fluorophores. For example, a significantly longer maturation time of mCherry in comparison to eGFP is consistent with eGFP + cells expressing *mCherry* transcripts, but not mature protein[54,55]. We mitigated the effect of this variability on our measurements of enhancer activity by quantifying fluorescence over the entire population of cells and using the heart as an endogenous control (Fig. 1d).

We next generated a transgenic mouse line in which a hZRS^446T>A variant allele identified in a family with preaxial polydactyly drives *mCherry* (hZRS^446T>A-*mCherry)*[56]. The 446T>A variant is hypothesised to create a de novo activator binding site, but its effect on in vivo hZRS enhancer activity is unknown. In contrast, the well-characterised 404G>A variant disrupts a repressor binding site and causes ectopic reporter expression in the anterior limb bud mesenchyme[41,47]. To visualise hZRS^404G>A and hZRS^446T>A variant allele activities simultaneously, we bred these mouse lines, each containing two copies of the respective transgene to generate two-colour transgenic embryos (Supplementary Fig. 1a, b). In this hZRS^446T>A-*mCherry*/hZRS^404G>A-e*GFP* transgenic embryos, mCherry and eGFP were detected in a highly

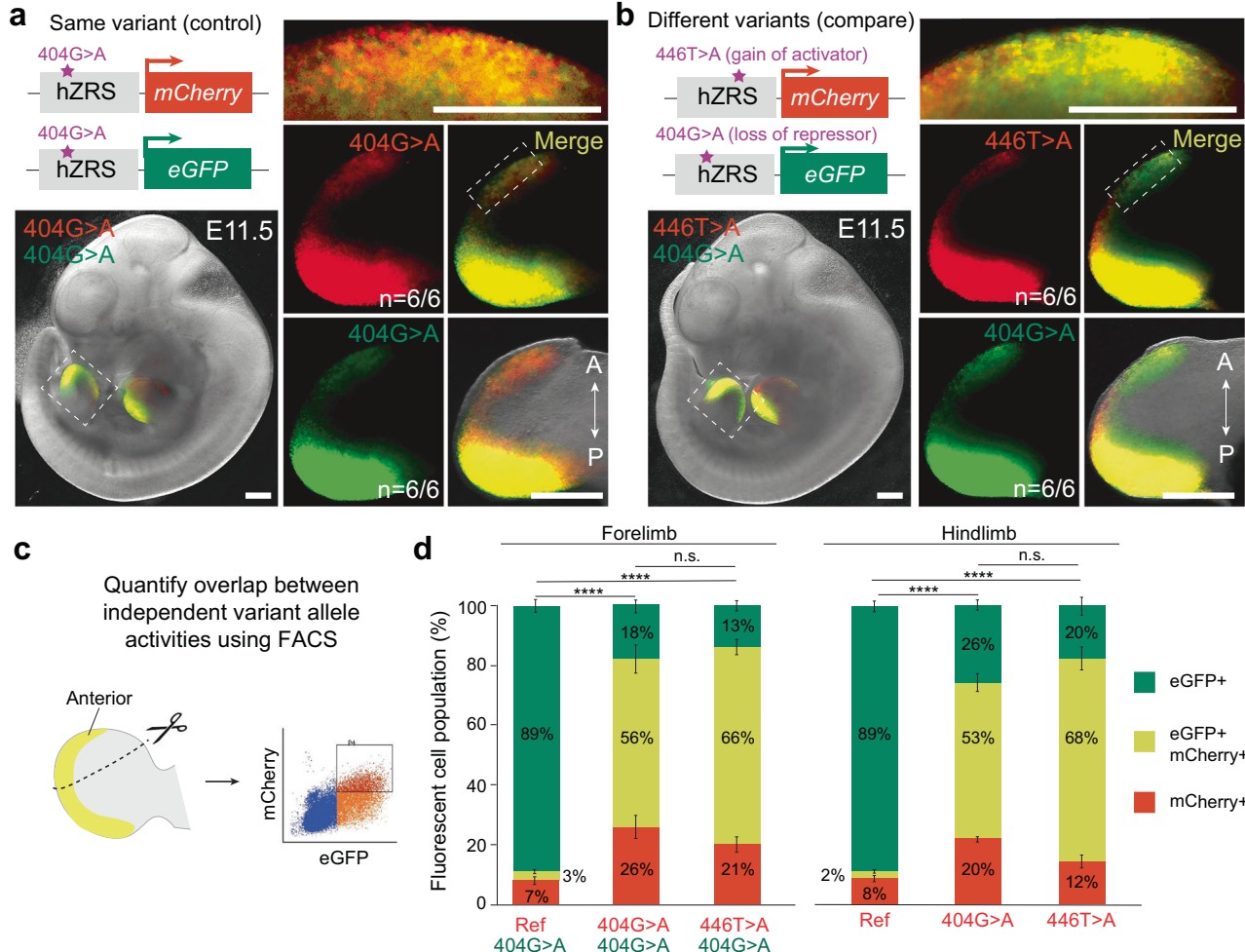

**Fig. 2 | Comparison of the effects of independent human-disease-linked variants in the hZRS enhancer. a** Sample image of an E11.5 hZRS[404G>A]-*mCherry*/hZRS[404G>A]-*eGFP* embryo. Panels on the right show high-resolution images of the hindlimb bud and anterior domain (above), as marked by outlined boxes. **b** Representative image of hZRS[446T>A]-*mCherry*/hZRS[404G>A]-*eGFP* embryo at E11.5. Panels on the right show high-resolution images of the hindlimb bud and its anterior domain (above), as marked by outlined boxes. **c** FACS-based quantification of the overlap between mCherry- and eGFP-expressing cells in the anterior domain of limb buds. **d** Plots depicting population distribution of eGFP +, eGFP + mCherry +, and mCherry + cells in the anterior domain of fore- and hindlimbs. Each

genotype contains at least three independent biological replicates. Genotypes of dual-enSERT-1 embryos are labelled along the *x*-axis and coloured according to the downstream reporter gene. Fisher's exact tests. hZRS[Ref]-*mCherry*/hZRS[404G>A]-*eGFP* vs. hZRS[404G>A]-*mCherry*/hZRS[404G>A]-*eGFP*: Forelimb, $P < 2.2E-16$; Hindlimb, $P < 2.2E-16$. hZRS[Ref]-*mCherry*/hZRS[404G>A]-*eGFP* vs. hZRS[446T>A]-*mCherry*/hZRS[404G>A]-*eGFP*: Forelimb, $P < 2.2E-16$; Hindlimb, $P < 2.2E-16$. hZRS[404G>A]-*mCherry*/hZRS[404G>A]-*eGFP* vs. hZRS[446T>A]-*mCherry*/hZRS[404G>A]-*eGFP*: Forelimb, $P = ns$; Hindlimb, $P = ns$. Data represented as mean ± SEM for plots. All scale bars, 500 µm. Source data are provided as a Source Data file.

overlapping pattern in the ZPA and anterior limb bud mesenchyme at E11.5 (ZPA, *P* = ns; Anterior, *P* = ns; Fig. 2b and Supplementary Fig. 2b).

The overlap in ectopic activity in the anterior limb bud mesenchyme was visually and quantitatively indistinguishable from the overlap observed in hZRS[404G>A]-*mCherry*/hZRS[404G>A]-*eGFP* transgenic embryos in which *eGFP* and *mCherry* were driven by the same hZRS[404G>A] variant allele (Fig. 2a, b and Supplementary Fig. 2a). We next examined this extent of overlap at cellular resolution in anterior cells from hZRS[446T>A]-*mCherry*/hZRS[404G>A]-*eGFP* transgenic embryos in which *eGFP* and *mCherry* were driven by different hZRS variants (Fig. 2b and Supplementary Fig. 3e). 66% (forelimb) to 68% (in hindlimb) of fluorescent cells in anterior limb buds were eGFP +/mCherry + (Fig. 2d). This fraction of double-positive anterior limb bud cells was not significantly different from the fraction of double-positive cells in hZRS[404G>A]-*mCherry*/hZRS[404G>A]-*eGFP* transgenic embryos. These data indicate that the 404G>A variant, which disrupts a repressor binding site, and the 446T>A variant, which creates an activator binding site,

both cause highly overlapping ectopic expression in the same population of anterior limb cells.

### Dual-enSERT-2 allows rapid comparison of enhancer allele activities in F0 mice

A limitation of the dual-enSERT-1 system is the ~ 6-month time required to obtain two-colour F2 embryos. This protracted timeframe limits the number of enhancer variants that can be rapidly tested in mice. To overcome this bottleneck, we constructed single transgenes containing *mCherry* and *eGFP* reporters driven by different enhancer alleles in divergent orientations. With this bicistronic system, henceforth referred to as dual-enSERT-2, an injection of a single construct would yield two-colour F0 embryos in as little as eleven days (Supplementary Fig. 4a). As a proof of principle, we placed the hZRS[ref] allele upstream of *mCherry* and hZRS[404G>A] variant allele upstream of *eGFP* all within the same construct. To prevent cross-activation between enhancer alleles and reporter genes, we separated two transgenes with three copies of

the well-characterised chicken $\beta$-globin insulator 5′-HS4. 5′-HS4 is widely used for its robust ability to block enhancer-promoter activation in the genome[57-59] and in the context of a zebrafish transgene[24]. To our surprise, separating the transgenes with three copies of 5′-HS4 completely failed to prevent enhancer cross-activation in single- and multiple-copy transgenes (Supplementary Fig. 4b, c and "Methods").

We thought to create a stronger synthetic insulator (SI) consisting of A2 (two copies), ALOXE3, and 5′-HS4 (two copies) insulators and placed it between two transgenes as well as in the vector backbone to prevent cross-activation between transgene copies[57,60,61] ("Methods"). In embryos with a single-copy transgene integration of hZRS$^{ref}$-mCherry/SI/hZRS$^{404G>A}$-eGFP at the H11 locus, mCherry fluorescence was restricted to the ZPA while eGFP fluorescence was also observed in the anterior limb bud (3/3 embryos; forelimb and hindlimb ZPA, $P$ = ns; 1.9-fold in anterior forelimb, $P$ = 0.0229; 2.1-fold in anterior hindlimb, $P$ = 0.00898; Supplementary Fig. 4d, e). These results mimic those of dual-enSERT-1 in which enhancer-reporters were placed on separate H11 alleles and indicate that the SI can prevent reporter cross-activation in the anterior limb bud. Furthermore, in transgenic embryos containing a single copy of the mouse ZRS (mZRS) driving mCherry and an enhancerless eGFP transgene (mZRS$^{ref}$-mCherry/SI/empty-eGFP), we observed only mCherry fluorescence in E11.5 limb buds, but no detectable eGFP fluorescence, indicating that the SI can fully insulate the two transgenes (Supplementary Fig. S4h, i)[62].

Interestingly, in embryos with multiple copies of a transgene separated by a synthetic insulator at the H11 locus, both anterior and posterior limb buds showed robust mCherry and eGFP expression in all examined embryos (8/8 embryos; Forelimb ZPA, $P$ = ns; Forelimb Anterior, $P$ = ns; Hindlimb ZPA, $P$ = ns; Hindlimb Anterior, $P$ = ns; Supplementary Fig. 4f, g). These results indicate that the hZRS can bypass the synthetic insulator in the context of a multi-copy transgene with multiple enhancer-reporters flanked by synthetic insulators (Supplementary Fig. 4j). Therefore, a synthetic insulator-based dual-enSERT-2 can discriminate between enhancer allele activities only if a single copy of the bicistronic transgene is integrated at the H11 landing site (Supplementary Fig. 4j).

To optimise the efficiency of dual-enSERT-2, we sought to maximise the number of single-copy integrants at the H11 landing site. Recent work in zebrafish and mice has shown that the addition of biotinylated nucleotides to the ends of donor DNA prevents concatemer formation during Cas9- mediated homology-directed repair[63,64] To test if the addition of biotin (B) results in preferential single-copy transgene integration at the H11 locus, we added biotinylated nucleotides to the ends of a linearised F0 dual-enSERT vector carrying the human reference and the 404G>A hZRS variant alleles ("Methods"). We injected this B-hZRS$^{ref}$-mCherry/SI/hZRS$^{404G>A}$-eGFP-B construct into mouse zygotes together with Cas9 ribonucleoproteins (RNPs) and imaged the F0 embryos eleven days later. mCherry fluorescence was restricted to the ZPA, while hZRS$^{404G>A}$-driven eGFP expression extended into the anterior limb bud in all embryos with transgene integration at the H11 locus (5/5 embryos, Forelimb ZPA, $P$ = ns; 1.4-fold difference in Forelimb Anterior, $P$ = 0.012; Hindlimb ZPA, $P$ = ns; 1.6-fold difference in Hindlimb Anterior, $P$ = 0.00037; Fig. 3a, b and Supplementary Data File 1). Moreover, genotyping confirmed that most transgenic embryos ($n$ = 5/6 embryos) contained a single-copy transgene at the H11 locus and lacked concatemers (Fig. 3a, b and Supplementary Data File 1). This result indicates that the addition of biotinylated nucleotides results in preferential single-copy integration of a bicistronic transgene and enables discrimination between enhancer allele activities in a straightforward manner.

### Dual-enSERT-2 detects the effects of different types of pathogenic enhancer variants
To test if the optimised dual-enSERT-2 can rapidly detect the quantitative effects of other disease-linked non-coding variants, we returned

to the autism- and intellectual disability-linked hs737/EBF3 locus (Supplementary Fig. 1c, d). We placed the hs737$^{ref}$ allele upstream of eGFP and the hs737$^{830G>A}$ variant allele upstream of mCherry. We then linearised, biotinylated, and injected the resulting B-hs737$^{ref}$-eGFP/SI/hs737$^{830G>A}$-mCherry-B construct into mouse zygotes with Cas9 RNPs. Eleven days later, live imaging revealed increased mCherry reporter expression in the dorsal forebrain (1.3-fold, $P$ = 0.044) while the ventral forebrain ($P$ = ns), midbrain ($P$ = ns), hindbrain ($P$ = ns), and neural tube ($P$ = ns) showed no difference between eGFP and mCherry reporter expression (Fig. 3c d, and Supplementary Data File 1). These results reproduce our earlier results using dual-enSERT-1, but in a much shorter time frame (11 days vs. 6 months).

We next asked if dual-enSERT-2 could be employed to study human disease variants that cause loss of enhancer activity. We focused on the previously characterised rare non-coding 350dupA mutation at the IRF6 locus that is linked to cleft lip formation[42]. 350dupA is a single A duplication at position 350 of the hs932 face enhancer of IRF6 (also known as MCS9.7). We created a bicistronic vector with the hs932 reference allele driving eGFP and the hs932$^{350dupA}$ variant allele driving mCherry separated by the synthetic insulator (B-hs932$^{ref}$-eGFP/SI/hs932$^{350dupA}$-mCherry-B; Fig. 3e). Imaging of transgenic mouse embryos at E11.5 revealed strong eGFP expression driven by the reference hs932 allele in the orofacial and limb ectoderm, as previously reported using LacZ-based transgenesis[42] (4/4 embryos, Fig. 3f and Supplementary Data File 1). By contrast, we found no mCherry fluorescence in all examined transgenic embryos, indicating a near complete loss of enhancer activity (10-fold in orofacial, $P$ = 0.00051; 8.5-fold in forelimb, $P$ = 0.00015; 7.7-fold in hindlimb, $P$ = 0.00022; Fig. 3e, f). Overall, 80% of all F0 mice with reporter integration at the H11 locus generated with biotinylated dual-enSERT-2 constructs contained only single-copy transgenes (Supplementary Data File 1). Altogether, these results demonstrate the utility of dual-enSERT-2 for rapidly detecting and quantifying gain- and loss-of activity for disease-linked enhancer variants in vivo.

### Scaled assessment of previously uncharacterised non-coding variants with dual-enSERT-2
We next sought to use dual-enSERT-2 to functionally screen previously uncharacterised human non-coding variants linked to congenital disease. We compiled a list of thousands of candidate pathogenic rare and common non-coding variants from patients with neurodevelopmental disorders (NDDs) identified by GWAS and WGS studies[17,26,65-68]. We intersected this list with previously validated in vivo human and mouse enhancers active at embryonic day E11.5. We further narrowed the list by focusing on enhancers affected by multiple independent non-coding variants or enhancers with a known NDD-linked target gene based on capture Hi-C data, reasoning these regions the least likely to be random mutations (Fig. 4a, Table 1 and "Methods")[69]. From this prioritisation, we chose fourteen rare SNVs and one common indel distributed between seven unique enhancers active in the forebrain, midbrain, hindbrain, or neural tube.

To efficiently test these prioritised variants, we generated compound variant alleles for each of the enhancers, in some instances up to three independent variants per enhancer (Fig. 4a). We placed these compound variant alleles upstream of eGFP and the corresponding reference enhancer alleles upstream of mCherry and compared them using dual-enSERT-2. We observed loss of enhancer activity in one enhancer (hs268), the gain of enhancer activity for one enhancer (hs1791), no detectable changes for four enhancers (Fig. 4 and Supplementary Fig. 5a–h), and one enhancer (hMM1518) was inactive in our assay possibly due to interspecies sequence divergence (Supplementary Fig. 5i). For example, the introduction of two rare, autism-linked 435G>T and 700C>T variants in the hs268 enhancer of MIR9-2 resulted in substantial loss of enhancer activity in the brain and neural

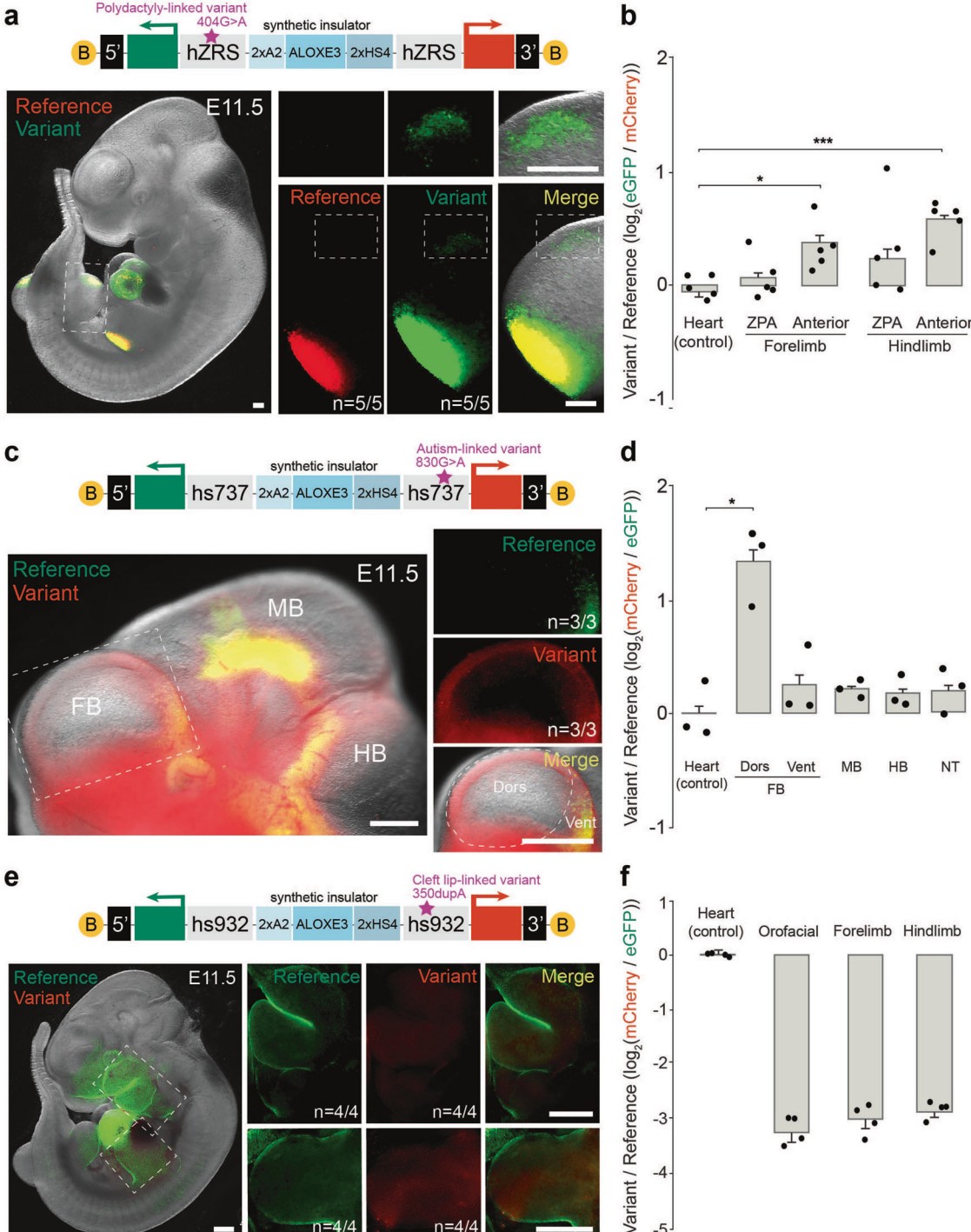

**Fig. 3 | An F0-based dual-enSERT-2 system for rapid testing of human enhancer variant activity. a** Fluorescent image of a B-hZRS<sup>ref</sup>-*mCherry*/SI/hZRS<sup>404G>A</sup>-*eGFP*-B whole embryo at E11.5 with close-up images of the hindlimb and anterior domain on right as marked by white dotted boxes. B, biotin. Scale bars, 250 μm. **b** Plots quantifying fold-change (log₂) difference in reporter intensity between variant and reference alleles by tissue in B-hZRS<sup>ref</sup>-*mCherry*/SI/hZRS<sup>404G>A</sup>-*eGFP*-B embryos. Data points represent independent biological replicates (*n* = 5 embryos). Two-sided paired *t* tests vs. Heart: Forelimb ZPA, *P* = ns; Forelimb Anterior, *P* = 0.0124; Hindlimb ZPA, *P* = ns; Hindlimb Anterior, *P* = 0.00037. Data represented as mean ± SEM. **c** Sample fluorescent image of a B-hs737<sup>ref</sup>-*eGFP*/SI/hs737<sup>830G>A</sup>-*mCherry*-B embryo at E11.5 with high-resolution image of forebrain on right. The white outline highlights ectopic areas with the gain in enhancer activity (red channel). B, biotin. Scale bars, 500 μm. **d** Plot depicting quantification of fold-change (log₂) difference in variant-reference reporter intensity by tissue in

B-hs737<sup>ref</sup>-*eGFP*/SI/hs737<sup>830G>A</sup>-*mCherry*-B embryos. Data points represent independent biological replicates (*n* = 3 embryos). Two-sided paired t-tests vs. Heart: Dorsal (Dors) Forebrain, *P* = 0.044; Ventral (Vent) Forebrain (FB), *P* = ns; Midbrain (MB), *P* = ns; Hindbrain (HB), *P* = ns; Neural Tube (NT), *P* = ns. Data represented as mean ± SEM. **e** Representative image of a B-hs932<sup>350dupA</sup>-*mCherry*/SI/hs932<sup>ref</sup>-*eGFP*-B embryo, with expanded views on the right to highlight the orofacial region and limbs at E11.5, as marked by white dotted boxes. B, biotin. Scale bars, 500 μm. **f** Quantitative plot for fold-change (log₂) difference in variant-reference fluorescent reporter intensity for heart, orofacial region, and limbs from B-hs932<sup>ref</sup>-*eGFP*/SI/hs932<sup>350dupA</sup>-*mCherry*-B embryos. Data points represent four biological replicates (*n* = 4 embryos). Two-sided paired *t* tests vs. Heart: Orofacial, *P* = 0.00051, Forelimb, *P* = 0.00015; Hindlimb, *P* = 0.00022. Data represented as mean ± SEM. Source data are provided as a Source Data file.

tube (5/5 embryos; 4.4-fold in the forebrain, $P = 0.001$; 3.2-fold in the midbrain, $P = 0.0093$; 3.3-fold in the hindbrain, $P = 0.0065$; 3.4-fold in the neural tube, $P = 0.0056$; Fig. 4b, c). 435G>T and 700C>T variants disrupt evolutionary conserved putative binding sites for neuronally-expressed transcription factors TBX/TBR2 and BCL11A, respectively (Fig. 4d). Conversely, a variant allele of the hs1791 midbrain enhancer of *OTX2* containing a common short tandem repeat (STR) polymorphism linked to autism (rs1880784044) resulted in an almost two-fold increase in midbrain enhancer activity (4/4 embryos; $P = 0.011$; Fig. 4e–g)[68]. These results indicate that dual-enSERT-2 can be used for rapid functional screening of non-coding variants linked to congenital disorders.

### Pathogenic enhancer variant activity at single-cell resolution

We next asked whether ectopic activity caused by gain-of-function enhancer variants can be quantitatively assigned to specific cell types in vivo using dual-enSERT. Such information, coupled with gene expression profiling can potentially reveal which genetic pathways lead to ectopic gene expression upon enhancer misregulation. We focused on the pathogenic hZRS$^{404G>A}$ variant allele for which the mechanism of ectopic *Shh* expression in the limb bud is not known (Fig. 1c)[41]. We performed two separate single-cell RNA-sequencing (scRNA-seq) experiments on dissected E11.5 hindlimb buds from F2 dual-enSERT-1 and F0 dual-enSERT-2 embryos, respectively (Figs. 1c, 3b, 5a and Supplementary Fig. 6a). In both transgenic embryos, hZRS$^{ref}$ allele drove *mCherry*, while hZRS$^{404G>A}$ allele drove *eGFP*. To decrease reporter gene dropout[70,71], we adopted a nested PCR strategy to amplify *mCherry* and *eGFP* transcripts in our barcoded libraries (Methods and Fig. 5a)[72]. We processed scRNA-seq datasets generated from dual-enSERT-1 and dual-enSERT-2 hindlimb buds independently (Supplementary Fig. 6a). Supervised clustering of dual-enSERT-1 hindlimb produced thirteen distinct cell types, including a large mesenchymal cluster defined by the specific expression of well-known marker genes (Supplementary Fig. 6b and Supplementary Data File 3)[73,74]. We recovered every cell type in dual-enSERT-2 hindlimb, except for proximal cells which we excluded during hindlimb dissection for dual-enSERT-2 (Supplementary Fig. 6b). Having shown the reproducibility between dual-enSERT-1 and -2, we combined and integrated the two scRNA-seq datasets together to obtain ~ 21,000 cells and maximise statistical power for downstream analyses (Fig. 5b and Supplementary Fig. 6c).

We first asked in which cell types are the reference and variant hZRS alleles active based on reporter gene expression. The strongest cluster of *mCherry* expression, driven by the reference hZRS allele, was distal posterior mesenchyme, matching the expression of its target gene *Shh* (Fig. 5c–e). We also detected *mCherry* expression in immune cells where *Shh* is not expressed, possibly due to *hsp68* promoter activity[20]. By contrast, strong levels of *eGFP* expression driven by the hZRS$^{404G>A}$ allele were also detected in distal middle, anterior, and proximal anterior mesenchymal cells, matching the distribution of ectopic eGFP fluorescence in live embryos (Fig. 5c, d). These cell-type-specific expression patterns were consistent between F2 dual-enSERT-1 and F0 dual-enSERT-2 datasets (Supplementary Fig. 6d).

We next examined gene expression in cell subpopulations in which hZRS is normally active (*mCherry* +/*eGFP* +), ectopically active (*eGFP* +/*mCherry*-) and inactive (*mCherry*-/*eGFP*-). We performed unbiased differential gene expression analysis between these cell subpopulations to identify candidate genetic pathways linked to normal and ectopic *Shh* expression. With only thirty-six differentially expressed genes (FDR < 0.05, $\log_2$FC > $\pm$ 1.5), both normal and ectopic hZRS domains showed strong similarity in their transcriptional profiles, including enrichment of known mesenchymal transcription factors such as *Msx1*, *Lhx2*, *Lhx9*, *Twist1* and others (Fig. 5e). This is consistent with the fact that many transcription factors specifying mesenchymal fate are expressed in the entire "progress zone" beneath

the apical ectodermal ridge (AER) which includes ectopic anterior and normal posterior domains of hZRS activity[75,76]. By contrast, inactive cells expressed chondrocyte-specifying transcription factors like *Shox2* and *Sox9* (Fig. 5e)[77,78]. Differential gene expression analysis between three clusters identified only a few genes, such as *Asb4* that were specifically expressed in ectopic *eGFP*+ /*mCherry*- cells.

To increase the sensitivity of the analysis, we subset cells expressing *mCherry* and/or *eGFP* and re-clustered them (Fig. 5f). We found that cells clustered into those expressing both *mCherry* and *eGFP* (normal domain, Clusters 1 and 2) and those only expressing *eGFP* (ectopic domain, Cluster 3) (Fig. 5f, g). To identify genes specifically upregulated in *eGFP*-expressing cells, we performed differential analysis between Cluster 3 and all other clusters. We identified several anterior-biased genes, including *Asb4*, *Pax9*, and *Hpse2* (Fig. 5h), which match their expression patterns by in situ hybridisation experiments of the limb bud. Taken together, these results implicate candidate pathways in ectopic hZRS activity and highlight how dual-enSERT enables the capture of variant allele-labelled cells at cellular resolution.

## Discussion

Realising the functional and therapeutic potential promised by large-scale human genomics studies depends on our ability to test the in vivo effects of candidate non-coding variants linked to human disease. However, no method currently exists for direct comparative testing of enhancer variant activity in a live mammal. In this study, we directly addressed this unmet need by developing dual-enSERT, a rapid, quantitative, and cost-effective method for simultaneous comparison of human enhancer allele activities in live mice. Dual-enSERT can be used to study rare and common non-coding variants and is compatible with a wide range of congenital disorders and organ systems. We also show that dual-enSERT can be easily combined with single-cell technologies to characterise gene expression in cells where an enhancer is ectopically active.

Our functional screening of previously uncharacterised candidate variants from WGS and GWAS shows reproducible effects on the in vivo enhancer activity of two out of seven tested enhancers. These effects were detected from just a few embryos by using quantifiable fluorophores and both enhancer alleles placed on one transgene, eliminating the impact of mosaicism seen in enSERT and traditional LacZ assays, which require large numbers of embryos to detect such effects[17,20,79,80]. The remaining five of seven variant enhancers did not change their activity in transgenic mice upon the introduction of these variants. It is possible that the variants that did not affect enhancer activity at E11.5 might have an impact during different developmental stages. Alternatively, these variants may be benign incidental observations in cases whose underlying disorder is caused by other genetic and/or environmental factors, which is often the case in human genetics studies[20,81].

While developing dual-enSERT-2, we found that three copies of the widely used 5′-HS4 chicken β-globin insulator, each of which contains multiple CTCF sites[24,57,58], unexpectedly failed to block communication between the hZRS enhancer and the *hsp68* minimal promoter (Supplementary Fig. 4b, c). These results support previous observations that insulator function can be locus- or enhancer-specific[82–84]. There is also evidence that insulator function is transcription factor-specific, i.e., some insulators work with one type of enhancer but not with another[83,84]. Therefore, caution should be taken when using individual insulators to protect transgenes in genome engineering applications.

Our synthetic insulator, created by fusing multiple copies of three of the most well-studied vertebrate insulators, A2, ALOXE3, and 5′-HS4, effectively blocked promoter interactions for five enhancers in the context of a transgene. We speculate the ability of SI to efficiently block enhancer-promoter interactions may derive from combining different mechanisms of insulation: A2 and 5′-HS4 depend on CTCF

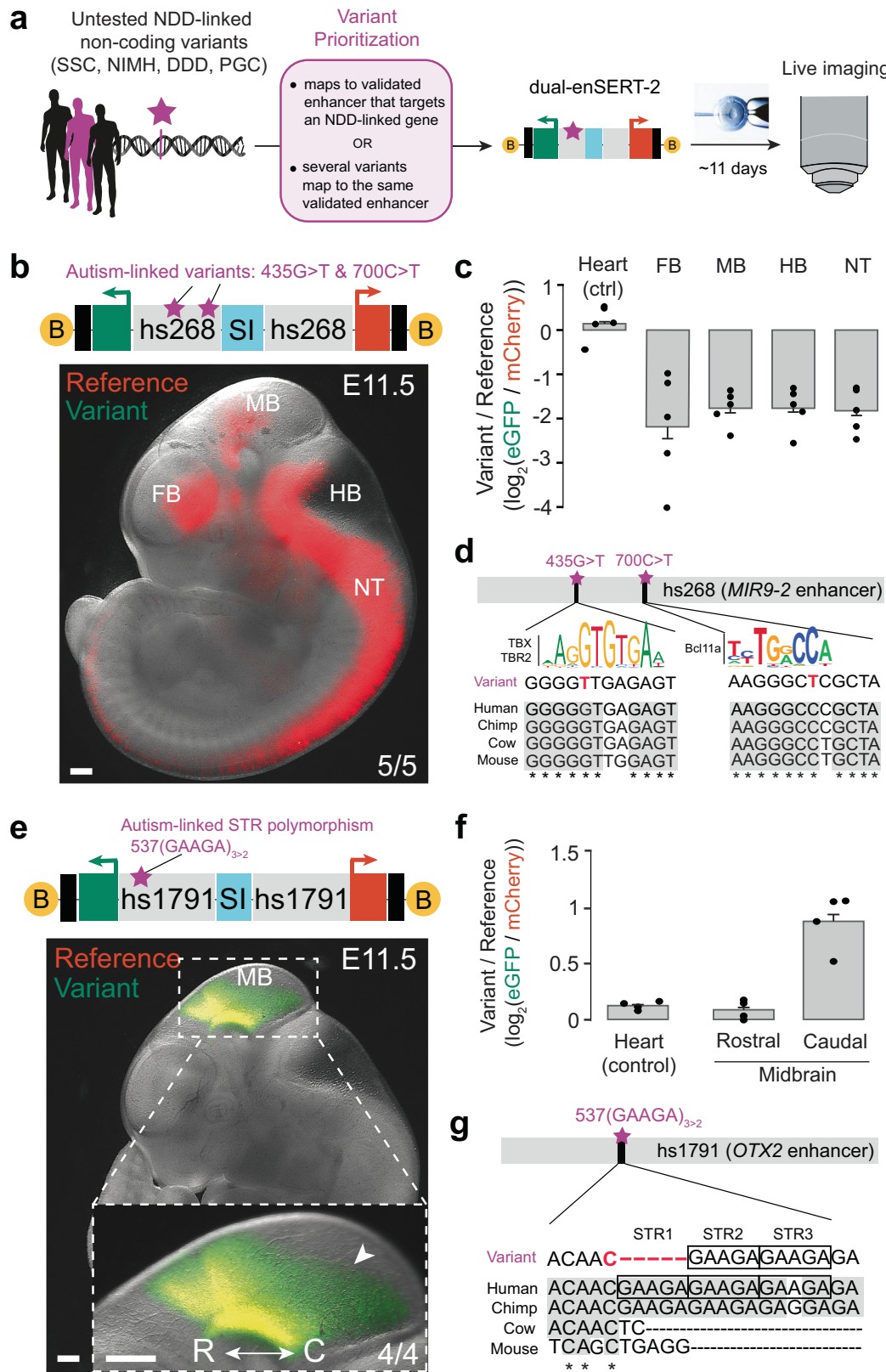

while ALOXE3 relies on two B-boxes that recruit RNA polymerase III[61,85,86]. This SI has the potential to work in a wider number of genomic contexts and applications.

Beyond studying disease-linked enhancer variants, dual-enSERT can be used for other applications. Sequence divergence in enhancers is hypothesised to be a major driver of morphological and functional evolution[87,88]; however, pinpointing and functionally testing the causal regulatory regions has been challenging. With dual-enSERT, the activity of candidate evolution-driving enhancers can be directly compared to a reference enhancer in the same animal to detect any functional changes. Dual-enSERT also provides a more time-effective and quantitative method for in vivo mutagenesis of enhancers that

**Fig. 4 | In vivo testing of uncharacterised variants from patients with autism spectrum disorder. a** Schematic depicting the identification and testing of uncharacterised human enhancer variants linked to neurodevelopmental disorders (NDDs). DDD, Deciphering Developmental Disorders; NIMH, National Institute of Mental Health cohort; PGC, Psychiatric Genomics Consortium; SSC, Simons Simplex Collection. **b** Fluorescent image of B-hs268ref-*mCherry*/SI/hs268var-*eGFP*-B whole embryo at E11.5. B, biotin. Scale bars, 500 µm. **c** Plots quantifying fold-change (log2) difference in reporter intensity between variant and reference alleles by tissue in B-hs268ref-*mCherry*/SI/hs268var-*eGFP*-B embryos. Data points represent independent biological replicates ($n = 5$ embryos). Two-sided paired $t$ tests vs. Heart: Forebrain (FB), $P = 0.0011$; Midbrain (MB), $P = 0.0093$; Hindbrain (HB), $P = 0.0065$; Neural Tube (NT), $P = 0.0056$. Data represented as mean ± SEM. **d** Putative TF motifs within the hs268 enhancer of *MIR9-2* that overlap with the loss-of-function patient SNVs at the 435 and 700 nucleotide positions with conservation. **e** Fluorescent image of B-hs1791ref-*mCherry*/SI/hs1791var-*eGFP*-B whole embryo and high-resolution inset of the midbrain (MB) at E11.5. B, biotin; C, caudal; R, rostral. Scale bars, 500 µm. **f** Plots quantifying fold-change (log2) difference in reporter intensity between variant and reference alleles by tissue in B-hs1791ref-*mCherry*/SI/hs1791var-*eGFP*-B embryos. Data points represent independent biological replicates ($n = 4$ embryos). Two-sided paired $t$ tests vs. Heart: Rostral Midbrain, $P = $ ns; Caudal Midbrain, $P = 0.011$. Data represented as mean ± SEM. **g** A common polymorphism variant at the 537-nucleotide position causes a reduction in short tandem repeat copy-number of GAAGA in the hs1791 enhancer of *OTX2*. Human silhouette and microscope cartoons reproduced courtesy of Zane Mitrevica and Augustin Carpaneto, respectively (http://sci-draw.io). Source data are provided as a Source Data file.

**Table 1 | Summary of dual-enSERT-2 results for previously uncharacterised human enhancer variants**

| Variant Location (hg38) | dbSNP ID | Associated Disorder | Enhancer (Putative Target Gene(s)) | Normal Enhancer Activity | Reproducible Changes in Enhancer Activity | Reference |
|---|---|---|---|---|---|---|
| chr1:213425381T>C, chr1:213425533A>C | – – | Autism spectrum disorder | hs204 (*PTPN14*) | Forebrain | No change | Shin et al.[26] |
| chr5:88396771G>T, chr5:88397035C>T | rs576375513, – | Autism spectrum disorder | hs268 (*MIR9-2*) | Forebrain, Midbrain, Hindbrain, and Neural Tube | Loss of activity in the Forebrain, Midbrain, Hindbrain and Neural Tube | Shin et al.[26] |
| chr19:30843449G>C, chr19:30843509G>A | – – | Neurodevelopmental disorder | hs430 (-) | Midbrain | No change | Short et al.[67] |
| chr3:180462583T>C, chr3:180462704A>G | rs189450851, – | Neurodevelopmental disorder | hs655 (*RNF220, ERI3, DMAP1*) | Hindbrain | No change | Short et al.[67] |
| chr3:147847042T>C, chr3:147847133T>C, chr3:147847216T>C | –, –, rs1166255572 | Autism spectrum disorder | hs1573 (-) | Forebrain, Hindbrain | No change | Shin et al.[6] |
| chr14:57007962CGAAGA>C | rs1880784044 | Autism spectrum disorder | hs1791 (*OTX2*) | Midbrain | Gain of activity in the Midbrain | Grove et al.[68] |
| chr15:66774248C>T, chr15:66774318C>T, chr15:66774321C>T | –, –, rs546228412 | Autism spectrum disorder | hMM1518 (*SMAD6, SCARLETLTR*) | Not Active | No change | Zhou et al.[97] |

allows a whole embryo readout of changes in enhancer activity. Another potential application of dual-enSERT is the genetic labelling of specific cell populations. It is often difficult to isolate a desired cell population with a single genetic driver but using an intersectional strategy with two fluorescent reporters driven by different enhancers can enable labelling and isolation of smaller cellular populations[89–91].

When testing non-mouse enhancers with dual-enSERT, we cannot rule out the potential effects caused by *trans*-regulatory divergence between different species, especially if no detectable changes in variant enhancer activity are observed[20]. In addition, dual-enSERT has a limited throughput which only allows testing two enhancer alleles per construct. Nevertheless, dual-enSERT provides a valuable addition to high-throughput methods such as MPRAs, as changes in enhancer activities are detected in vivo in whole live mice and in a native chromosomal context. In summary, our work demonstrates the power of mouse transgenesis by enabling rapid and quantitative comparative in vivo testing of disease-linked variants for the interpretation of new human genetics findings.

## Methods

### Ethics statement

This research complies with all relevant ethical regulations. All animal procedures, including those related to the generation of transgenic animals, were conducted in accordance with the guidelines of the National Institutes of Health (NIH) and approved by the Institutional Animal Care and Use Committee at the University of California, Irvine under protocol no. AUP-23-005.

### Cloning of dual-enSERT constructs

**Dual-enSERT-1 plasmid construction.** To create dual-enSERT-1 constructs, we used PCR4-Hsp68::lacZ-H11 plasmid[20] (Addgene #139099). We replaced *lacZ* with *eGFP* or *mCherry* fluorescent reporters using Gibson cloning[92]. The resulting constructs contained *eGFP* (PCR4-Hsp68::eGFP-H11) or *mCherry* (PCR4-Hsp68::mCherry-H11) fluorescent reporter genes under the control of the *hsp68* minimal promoter and homology arms targeting the H11 locus. Each tested enhancer sequence was cloned into the corresponding dual-enSERT-1 vector using NotI digestion followed by Gibson assembly (NEB, E2611)[38,92]. Reference allele sequences for hZRS and hs737 enhancers were obtained via PCR cloning from human genomic DNA (Promega, G304A). Primers used for each enhancer sequence are outlined in Supplementary Data File 2. All PCR cloning was performed using Q5 High-Fidelity Polymerase (NEB, M0491) or KOD polymerase (Toyobo, #KMM-201). Variant allele sequences for hZRS404G>A, hZRS446T>A, and hs737830G>A were synthesised as gBlocks (Integrated DNA Technologies (IDT)).

**Dual-enSERT-2 plasmid construction.** The bicistronic plasmid was designed as a modified version of the fluorescent dual-enSERT-1 plasmid. The cHS4 insulator sequence was cloned from genomic chicken DNA (Zyagen, GC-314) using the primers from Bhatia et al. To obtain three copies, linker sequences (fragments of the LacZ gene) were added as overhangs to the primers. The A2 insulator was synthesised as a gBlock using the sequence reported by Liu et al.[61] while ALOXE3 was cloned from human genomic DNA using primers reported by Raab

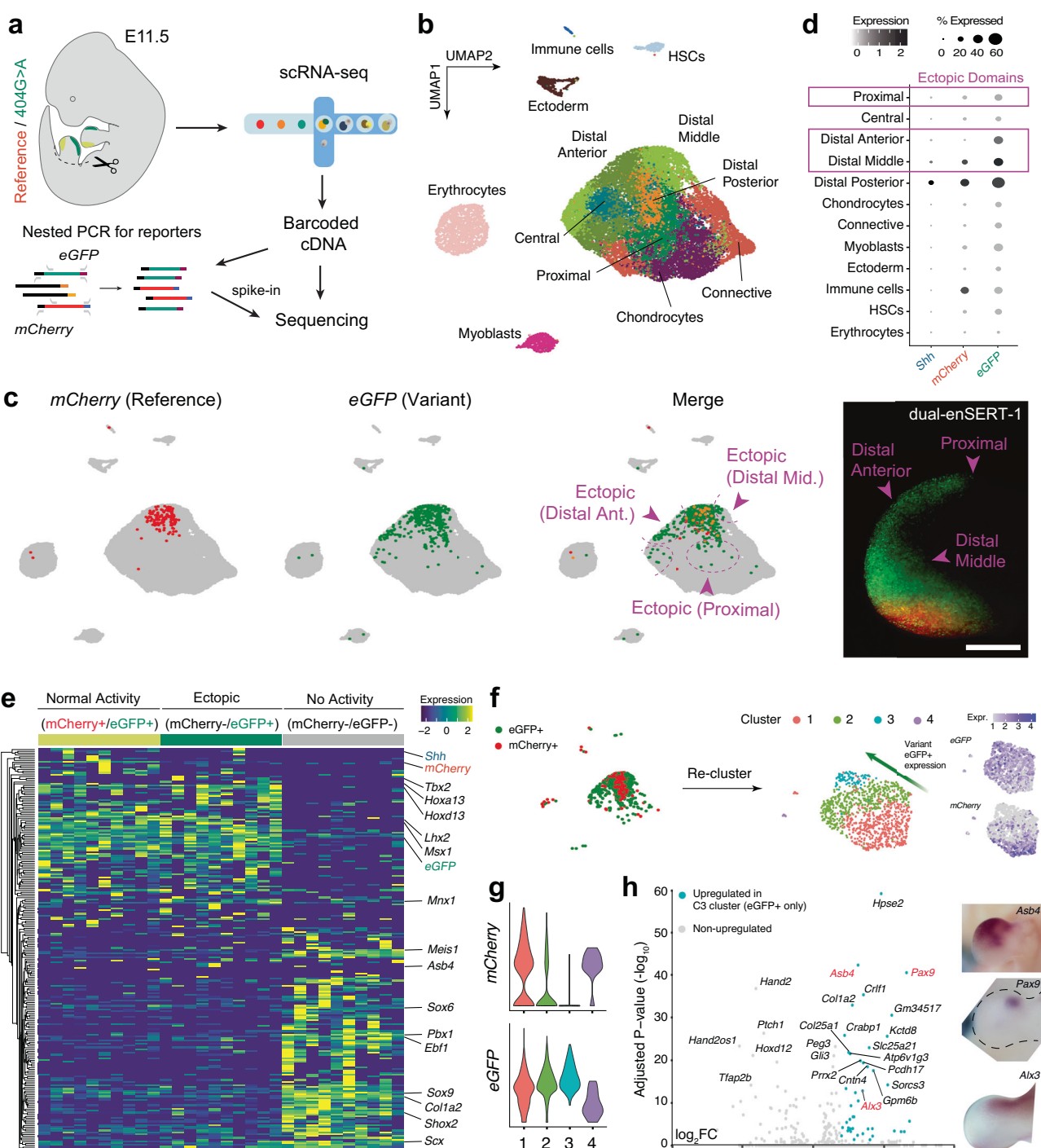

**Fig. 5 | Characterisation of pathogenic enhancer variant activity at single-cell resolution. a** Single-cell transcriptomic profiling of E11.5 hindlimbs from transgenic mice in which hZRS[ref] drives *mCherry* and hZRS[404G>A] drives *eGFP*. A nested PCR strategy was used to amplify *mCherry* and *eGFP* reporter transcripts (Methods). **b** Integrated UMAP plot showing ~21,0000 cells clustered by cell type. Mesenchymal clusters are defined by the expression of spatially-defined genes. HSCs, hematopoietic stem cells. **c** Feature plots showing *mCherry* and *eGFP* expression with overlapping cells marked in yellow. Areas of ectopic *eGFP* expression are highlighted in magenta with an accompanying fluorescent image on the right. Data derived from one dual-enSERT-1 and one dual-enSERT-2 replicate. Scale bar, 500 μm. **d** Dot plot quantifying percentage and normalised expression of cells expressing e*GFP*, *mCherry*, and *Shh* within each cluster. Clusters with ectopic *eGFP*

expression are highlighted. **e** Heatmap of differentially expressed genes between mCherry + (normal hZRS activity), mCherry-/eGFP + (ectopic hZRS activity), and mCherry-/eGFP- (inactive ZRS) cell subpopulations. Unsupervised hierarchical clustering of genes on the left; select marker genes on the right. **f** UMAP plots of reclustered mCherry + and eGFP + cells with accompanying feature plots depicting their expression. **g** Violin plots quantifying *mCherry* and *eGFP* expression across clusters. **h** Volcano plot depicts differential gene expression between Cluster 3 and the other clusters. Genes upregulated (Adjusted *P*-value < 0.05 and log₂FC > 2) in Cluster 3 are coloured in teal. Non-parametric Wilcoxon-rank sums test. The spatial distribution of representative marker genes in E11.5 limb buds is shown on the right. Images reproduced with permission from the Embrys database (http://embrys.jp).

et al.[60]. Fusion PCR was performed to obtain the final synthetic insulator fragment consisting of two copies of A2, one copy of ALOXE3, and two copies of cHS4. Enhancer-*hsp68p*-reporter sequences were then PCR amplified from dual-enSERT-1 plasmids.

To streamline the cloning of different enhancers into dual-enSERT-2 plasmids, NotI and AgeI restriction digestion sites were added to the outside of the two enhancer sites via PCR. Dual-enSERT-2 plasmid was digested with NotI (NEB, R3189) and AgeI (NEB, R3552) to create an empty vector without enhancers. Then, plasmids were assembled using a four-fragment Gibson-based assembly of (i) empty dual-enSERT-2 backbone, (ii) reference enhancer allele, (iii) variant reference allele, and (iv) synthetic insulator. The hZRS and hs737 reference and variant allele sequences were PCR amplified from dual-enSERT-1 plasmids. All other reference enhancer alleles were cloned from human genomic DNA using primers listed in Supplementary Data File 2. When sequence complexity was sufficiently low for in vitro synthesis, variant enhancer alleles were synthesised as gBlocks (IDT). If unable to be synthesised, custom primers with overhangs containing the selected variants were designed and used to clone separate fragments of the enhancer. Gibson-based cloning methods were then used to assemble the full-length variant enhancer alleles.

To enable linearisation of the dual-enSERT-2 donor plasmid, PauI (also known as BssHII) sites were added to the outside of the H11 homology arms via PCR. A dual-enSERT-2 plasmid (2 µg) was digested overnight with PauI (NEB, R0199) at 50 °C in rCutSmart Buffer. The following day, the reaction was inactivated by incubation at 65 °C for 15 min. To end-fill the 3′ overhangs with biotinylated nucleotides, Biotin-11-GTP (100 µM; Jena Bioscience, NU-971-BIOX) and biotin-11-CTP (100 µM; Jena Bioscience, NU-831-BIOX) were added to the reaction with T4 DNA Polymerase (1 unit; NEB, M0203) and incubated at 12 °C for 15 min. The reaction was stopped by the addition of EDTA (100 mM), and heat inactivated at 75 °C for 20 min. Biotinylation of fragments was confirmed by pull-down with streptavidin T1 Dynabeads (Thermo Fisher Scientific, cat. no. 65601).

For all constructs in this study, restriction digestion with SacI, Eco72I, NotI and/or AgeI (Thermo Fisher, FastDigest), Sanger and/or whole-plasmid (Plasmidsaurus) sequencing were performed to ensure the integrity of the vector and enhancer sequences before zygote microinjection. See Supplementary Table 1 and Supplementary Fig. 7 for complete details of all plasmids created and used in this study, of which the construct backbones are available at Addgene (#211940, #211941 and #211942).

## Assessment of 5′-HS4 insulator for dual-enSERT-2

We first tested whether three copies of the previously characterised chicken β-globin insulator, 5′-HS4, could prevent the cross-activation of two enhancer-reporter transgenes. 5′-HS4 is widely used for its robust ability to block enhancer-promoter activation in the genome[57–59] and in the context of a zebrafish transgene[24]. We additionally placed three copies of the 5′-HS4 insulator (3xHS4) into the plasmid backbone to prevent cross-activation between different copies of the transgene in the event of multi-copy integrations at the H11 landing site[20] (Supplementary Fig. 4a). We injected the resulting hZRS^ref-*mCherry*/3xHS4/hZRS^404G>A-*eGFP* bicistronic construct into mouse zygotes and collected transgenic embryos at E11.5. We detected robust mCherry and eGFP expression in anterior cells in all examined transgenic embryos, indicating that the variant hZRS allele can simultaneously activate *eGFP* and *mCherry* reporter genes in this transgene context (5/5 of embryos with a single-copy transgene integration at H11; 11/11 of embryos with multi-copy transgene integration at H11, Supplementary Fig. 4b, c and Supplementary Data File 1). Quantitatively, we found no difference in fluorescent intensity across the anterior and ZPA regions of the fore- and hindlimb (Forelimb ZPA, *P* = ns; Forelimb Anterior, *P* = ns; Hindlimb ZPA, *P* = ns; Hindlimb

Anterior, *P* = ns; Supplementary Fig. 4b, c). These results show that three copies of the 5′-HS4 insulator are insufficient to insulate the variant hZRS allele from cross-activating the other reporter gene (Supplementary Fig. 4j).

## Transgenic mouse generation

All transgenic mice in this study were generated using a CRISPR/Cas9 microinjection protocol, as previously described[20,93]. Briefly, a mix of (i) Cas9 protein (final concentration of 20 ng/µl; IDT Cat. No. 1074181), (ii) sgRNA (50 ng/µl) and (iii) circular donor plasmid (7 ng/µl) or linearised fragment (1 ng/µl) in injection buffer (10 mM Tris, pH 7.5; 0.1 mM EDTA) was injected into the pronucleus of FVB embryos. All donor plasmids or fragments were column purified using a PCR purification kit (Qiagen) and eluted into an injection buffer before injection. Female mice (CD-1 strain) were used as surrogate mothers. Superovulated female FVB mice (7–8 weeks old) were mated to FVB stud males, and fertilised embryos were collected from oviducts. The injected zygotes were cultured in M16 with amino acids at 37 °C under 5% CO2 for approximately 2 hr. Afterwards, zygotes were transferred into the uterus of pseudopregnant CD-1 females. F0 embryos were either brought to gestation (dual-enSERT-1) or collected at E11.5 (dual-enSERT-2).

## Mouse strains and embryo collection

Mice were maintained in standard housing conditions (temperature between 19–23 °C and humidity between 40–60%) on a reversed 12 h dark–light cycle with food and water provided *ad libitum*. Time of gestation was identified by the presence of vaginal sperm plugs, indicating E0.5. Pregnant dams were humanely euthanized, and E11.5 embryos were carefully removed under brightfield stereoscopes in ice-cold PBS (Cytiva, SH30256.01). Both sexes of embryos were presumed to be included. Yolk sacs or tail pieces were collected for genotyping. Successful integration events at the H11 locus were determined by PCR using primers described previously[20,38].

## Live fluorescent imaging and quantification

Embryos were imaged in ice-cold PBS in a small petri dish (Greiner Bio-One, #627102) atop a thin layer of 2% gel agarose (Fisher, BP160). Images were taken on a Zeiss V8 stereoscope using a monochromic camera (Axiocam 202, Zeiss), fibre optic light source (Zeiss, CL1500) LED fluorescent laser (X-Cite, Xylis), and fluorescent channels at 488 and 555 nm wavelengths. Single-channel images were merged using Zeiss BioLite software. Quantification of fluorescent reporter intensity was performed by importing the original.czi files into Fiji software[94]. Regions of interest were outlined on the variant-driven colour channel and then transferred to the reference-driven colour channel to measure the mean fluorescence intensities for each region. To account for intrinsic differences in fluorophore intensity and background, the mean fluorescence of embryo tissues with no enhancer activity was measured, averaged, and subtracted from the regions-of-interest fluorescent intensities. Because the hsp68 promoter causes leaky reporter activity in the heart[20], the heart was used as a negative control in two-sided, paired *t* tests to account for differences in fluorophore maturation time and half-life.

## Fluorescence-activated cell sorting

After imaging, the anterior portions of dual-enSERT-1 mouse limb buds carrying two hZRS reporter transgenes were carefully dissected under the fluorescent scope. Dissected regions from each embryo were pooled separately and then incubated with collagenase II (Gibco, #17101015, 0.2 µL at 100 u/µL) for 10 min at 700 rpm and 37 °C with trituration every 5 min with a P200 pipette. 450 µL of 10% FBS (Thermo Fisher, #A3840201) was added and dissociated cells were spun down at 500 × g for 5 min. Cells were resuspended in 200 µL of 0.04% BSA (Millipore Sigma, #A1595) and filtered using 40 µM P1000 Flowmi cell

filters (SP Bel-Art, #136800040). After gating with forebrain tissue as a negative control, mCherry +, eGFP +, and double-positive cell populations were quantified using a FACSAria Fusion Sorter (BD Biosciences). Fisher exact tests were performed between genotypes of double-positive, mCherry +, and eGFP + cells.

### RNA isolation and cDNA preparation
After fluorescent-activated cell sorting, 10 μL of RNAprotect (Qiagen, #76104) was immediately added to cells. RNA was isolated using the RNeasy Mini Kit (Qiagen, #74104) according to manufacturer instructions. cDNA libraries were constructed from isolated RNA with the ProtoScript® II First Strand cDNA Synthesis Kit (NEB, #E6560) using Oligo(d)T primers and following the manufacturer's recommended protocol.

### Quantitative PCR
Quantitative PCR was performed for mCherry and eGFP transgenes to quantify mRNA expression (from cDNA) and determine the copy-number of enhancer-reporter constructs in each dual-enSERT-1 mouse line (from genomic DNA). From mouse genomic DNA, mCherry or eGFP transgenes and endogenous control of known copy-number were amplified by primers and fluorescent probes (FAM or HEX, IDT), designed using PrimerQuest Tool (IDT) (Supplementary Data File 2). Reactions were performed in multiplex and carried out with Prime-Time Gene Expression Master Mix (IDT, 105577) in a C1000 Touch Thermal Cycler with a CFX96 Real-Time System module (Bio-Rad, 1845096). Cycle threshold values (Ct) for each amplicon were extracted from.zpcr files using CFX Maestro Software (Bio-Rad). Transgene copy numbers (normalised to endogenous control) were calculated using a modified $2^{-\Delta\Delta CT}$ method in which samples of unknown copy numbers were compared to positive control samples containing verified single transgene insertions. To quantify absolute mRNA expression from cDNA, mCherry or eGFP were normalised to the housekeeping gene Gapdh using the $2^{\Delta CT}$ method[95]. Plots were generated using GraphPad Prism, version 10.1.2.

### Prioritisation of disease-linked enhancer variants for dual-enSERT-2
To select previously uncharacterised human enhancer variants for testing with dual-enSERT-2, we extracted published rare and common variants identified from recent GWAS and WGS studies on patients with neurodevelopmental disorders[17,26,65–68]. The genomic locations of these variants were then intersected with the coordinates of mouse and human enhancers experimentally validated in vivo[36,37]. Enhancer variants were prioritised by either multiple independent variants mapping to the same enhancer or the enhancer interacting with the promoter of developmental disorder-lined gene based on published capture Hi-C data[69].

### Single-cell transcriptomics
Hindlimb buds from an hZRS^ref-mCherry/hZRS^404G>A-eGFP E11.5 (dual-enSERT-1) and B-hZRS^ref-mCherry/SI/hZRS^404G>A-eGFP-B (dual-enSERT-2) mouse embryo were dissected in ice-cold PBS and processed independently. A single-cell suspension was obtained using collagenase II, as described above for FACS. Dissociated cells were resuspended in 25 μL of 0.04% BSA before being quantified and inspected for viability using Trypan Blue (Bio-Rad, #1450013). Live cells were counted by hemocytometer (Bio-Rad, #1450011) and loaded at a concentration that would enable recovery of 10,000 nuclei by the Chromium Next GEM Chip G Single Cell Kit 3' Gene Expression Kit v3.1 (10X Genomics, cat no. 1000127). Captured cells were pair-end sequenced on an Illumina NovaSeq 6000 for ~50,000 reads per cell.

To amplify reporter transcripts, nested PCR for mCherry and eGFP transcripts was performed on the prepared cDNA library as described previously, with slight modifications[72]. The first PCR included a trimer mix of mCherry- and eGFP-specific forward primers (mCherry-1; eGFP-1) and an R1-targeting reverse primer (Supplementary Data File 2). After bead clean-up (CleanNA, CNGS-0050), a second PCR reaction was performed on the PCR product using a trimer mix targeting eGFP and mCherry (mCherry-2; eGFP-2) and the same R1 reverse primer (Supplementary Data File 2). The resulting DNA was then bead-purified in preparation for the final PCR using the i5:i7 indices from Chromium Next GEM Single Cell 3' GEM, Library & Gel Bead Kit v3.1 (10X Genomics, cat. no 1000128). All PCR reactions were performed using Q5 polymerase (NEB, M0491). The final indexed and purified DNA was spiked in for additional sequencing of 5000 reads per cell.

Fastq files were aligned to a modified mm39 genome assembly (Ensembl, GCA_000001635.9) that included mCherry and eGFP sequences and barcodes were counted to generate a count matrix using CellRanger (10x Genomics, Cell Ranger 3.1.0). Count matrix data were analysed using the Seurat R package, version 4[96]. A SeuratObject was created, and quality control was performed to exclude cells with greater than 5% mitochondrial DNA and with UMI counts between 2500 and 8000 per cell. Transcriptome data were normalised and scaled, and variably expressed genes (n = 2000) were utilised for principal component analysis. An elbow plot was produced to calculate the number of dimensions. Because the developing E11.5 limb bud is highly proliferative, cell cycle genes (S and G2M genes from Seurat) were regressed to enhance the detection of cell clusters based on spatial gene expression. Nearest neighbour, unsupervised clustering and UMAP analysis were then performed (dimensions = 13, resolution = 0.5). Cell types were identified from the resulting clusters using well-defined marker genes[73,74]. FeaturePlot was utilised to produce the overlaid plots of mCherry and eGFP expression. The DotPlot function was used to generate expression data by each cell cluster. To determine the topmost upregulated genes within the normal and ectopic hZRS domains, cells were selected for expression of the reporter transcripts: either mCherry > 2 (normal hZRS) or eGFP > 2 (variant hZRS), or inactive (remaining cells). Differential gene expression was defined by comparing cell populations using the FindMarkers function, and the heatmap was downsampled (n = 100) for easier visualisation. All barcode meta-data, including cell annotations and version of dual-enSERT, are reported in Supplementary Data File 3.

### Statistics and reproducibility
Statistical analyses were performed using R version 4.3.1 and Microsoft Excel version 16.79.1. Experimental parameters, including the number of embryos, statistical tests, and significance are reported in the text, figures and/or figure legends. The investigators were blinded to genotype for all imaging and quantification analyses. No statistical method was used to predetermine sample size, no data were excluded from the analyses, and no randomisation was performed. P-values less than 0.05 were considered significant. All bar graphs are shown as mean ± SEM. Raw sequencing data were analysed on the UCI high-performance computing cluster.

### Reporting summary
Further information on research design is available in the Nature Portfolio Reporting Summary linked to this article.

## Data availability
Processed and raw scRNA-seq data in this study have been deposited in the GEO database under accession code: GSE244244. Source data are provided in this paper.

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

## Acknowledgements

The authors would like to acknowledge the UCI Transgenic Mouse Facility for help in the generation of transgenic mouse lines, Elizabeth Pollina (Washington University at St. Louis) and Daniel Gillam (Harvard) for their technical assistance with the nested PCR protocol, and Zane Mitrevica and Augustin Carpaneto (via sci-draw.io) for open-access use of human silhouette and microscope cartoons, respectively. This work was made possible, in part, through access to the Genomics Research and Technology Hub Shared Resource of the Cancer Centre Support Grant (P30CA-062203) at the University of California, Irvine and NIH shared instrumentation grants 1S10RR025496-01, 1S10OD010794-01, and 1S10OD021718-01. The authors also thank Sabbi Lall (Life Science Editors) and Kvon lab members for their comments and suggestions on the manuscript. This work was supported by a National Institutes of Health grant R01HD115268 (to E.Z.K.), F30HD110233 (E.W.H.), T32NS082174 (E.W.H.), and T32GM008620 (E.W.H.) from the National Institutes of Health.

## Author contributions

E.W.H. and E.Z.K. conceived the project. E.W.H., T.A.L., S.H.J. and C.X.C. performed mouse transgenesis experiments, imaging and analysed the data. E.W.H. performed single-cell RNA-seq experiments and analysed the data. J.A.A. performed qPCR experiments and analysed the data. E.W.H. and E.Z.K. wrote the manuscript with input from all authors.

## Competing interests

The authors declare no competing interests.
