## [Peer Review file · Nature Communications]

Rapid and Quantitative Functional Interrogation of Human Enhancer Variant Activity in Live Mice

Corresponding Author: Dr Evgeny Kvon

Version 0:

Reviewer comments:

Reviewer #1

(Remarks to the Author)

The manuscript by Hollingsworth and colleagues creates a new fluorescence reporter system to test for enhancer activity in mouse embryos. Given the exponential increase in whole genome sequencing studies and the subsequent discoveries of thousands of non-coding variants, this reporter system is a valuable tool to test for function of those variants in vivo. The authors use 2 well defined enhancers, variants in which lead to polydactyly and associated with hypotonia, ataxia, and delayed development syndrome (HADDs) respectively. The authors also create a dual fluorescent reporter with multiple insulators cloned in. This allows for the testing two enhancers simultaneously, hence facilitating the testing of risk and non-risk alleles from the same construct, creating in effect a heterozygous enhancer.

Though the technology would be useful to many researchers using mouse model for enhancer validation, the real strength of this study is the ability to sort these cells and uncover the transcriptional and cellular consequence of the enhancer variants; But the authors fall short on this account. Thus, I am less enthusiastic about the study in its current form. The authors should delve deeper into their scRNA seq data and delineate specific genes which leads to such distinct ectopic expression from the various enhancer variants. Below are some comments which can help improve the study and make it really helpful not only for broad community of researchers studying enhancer function but more specifically researchers studying the ZRS enhancer and the EBF3 enhancer.

1) Though the authors perform single cell RNA seq study on the hindlimb bud of ZRSref-mCherry/ZRF404G>A-EGFP, the interpretation and analysis are very superficial (figure 3). The authors should ideally sort the mCherry+ cells, the GFP+ cells and double positive cells and redo the single cell assay. In absence of that the authors should at least subset out these cell (mCherry+, GFP+) populations from their current data and re-cluster them and then perform differential expression analysis. Their current analysis suggests there isn't much transcriptional differences between cells expression the wildtype and variant allele containing enhancers which I suspect is due to nature of the gene expression analysis.

2) It is also unclear how many genes were significantly differentially expressed between these subclusters even in the current analysis? The heatmap appears to be made with cherry-picked set of genes. Beyond the transcription factors and A-P axis patterning genes what other genes are differentially expressed? This section on scRNA-seq analysis needs to be expanded and better explained.

3) It is also unclear in dual-enSERT-1 experiments how many copies of both the wild type and mutant enhancer are integrated in the H11 locus? Copy number would play a major role in number of cells expressing the fluorescent molecules and hence affect the single cell data.

4) In figure 2 it is clear that the type of fluorescent molecule has an effect. In figure 2D (forelimb) 18% cells and 26% cells exclusively express 1 marker. It would be good for the authors to sort these single fluorescent labelled cells and do some sort of gene expression (even qPCR for selected markers including Shh and A-P axis marker) to see what if any are the transcriptional differences. This is an important piece of data to have for future studies when deciding on what fluorescent marker to pick.

5) The whole section "ZRS enhancer bypass three copies..." (line 269) should be moved to supplement and the authors should directly describe their dual-enSERT-2.0 study with synthetic insulators. They should direct the readers to the

supplementary section about the single insulator study and its ineffectiveness.

6)The authors should ideally perform scRNA-seq studies on their single copy dual-enSERT-2.2 embryos for hZRS404G>A as a comparison to their studies with dual-enSERT-1. This will allow for the authors to judge any off target cellular effects from their new construct and give confidence of its efficacy and specificity.

7)Line 224, reference 41 is in a different reference style.

Reviewer #2

(Remarks to the Author)

Hollingsworth et al. reported improved versions of enSERT—an efficient CRISPR/Cas9-mediated site-specific transgenic mouse enhancer assay (Kvon et al. Cell 2020). To validate cell type-specific spatial enhancer activity, they developed dual-enSERT using two different reporter genes. To test the idea of dual-enSERT, they examined the impact of a known variant on ZRS enhancer activity and were able to analyze the enhancer activities with or without variants in the same mouse embryo using FACS sorting and single-cell analysis. To analyze two enhancer activities in mice, they also developed a bicistronic reporter assay (dual-enSERT-2.2) and confirmed the reproducibility of ZRS, hs737, and hs932 enhancer activity with or without variants.

I understand the concept of this study and recognize that analyzing spatial enhancer activity is crucial. This dual-enSERT system can precisely analyze variants' impact in mice in vivo. However, my impression of this work is that it is more incremental. I can see several technologically significant findings, such as the synthetic insulator and the addition of biotinylated nucleotides to obtain a single copy transgene. However, as the author mentioned, compared to other high-throughput assays like MPRA, dual-enSERT is very low throughput and it's difficult to see significant advancements compared to conventional enSERT or LacZ transgenic assays.

Major points:

Given its broad readership, including non-mouse geneticists, I would like to see more data demonstrating a pathogenic mechanism with this system for publication in Nature Communications. Although the authors tested three variants associated with polydactyly, autism, and craniofacial malformations, the conventional LacZ transgenic assay has already been performed in previous work as cited in the manuscript, and it's difficult to see significant novel findings from this paper.

I really appreciate the idea of using this system to understand the mechanistic difference between 'gain of activator' versus 'loss of repressor' as shown in Figure 2. It's interesting that two different mutations cause the same polydactyly condition. The authors may want to conduct single-cell analysis for the 446T>A (gain of activator) variant with the same system as in Figure 3 and compare the cell subpopulations. As the author mentions on page 7, line 168, I am curious whether these independent mutations result in ectopic gene expression in the same or different cell populations of the limb bud.

As they have developed a rapid enhancer assay system, I would like them to explore uncharacterized variants identified from GWAS as an example. This might be out of scope, but it would certainly provide more valuable data and demonstrate the system's effective application.

Minor points:

1. Page 9, Line 206: The values for '50% (in forelimbs) to 56% (in hindlimbs)' are swapped between the forelimbs and hindlimbs.

2. When citing Fig. S3, please indicate parts A-G; it's hard to follow. Currently, there is no indication of A-G in the main text."

Version 1:

Reviewer comments:

Reviewer #1

(Remarks to the Author)

All my queries have been answered and this is a much improved manuscript. Its a very important and timely study and the reagents developed by the authors would be extremely useful to the human genetics community

Reviewer #2

(Remarks to the Author)

The authors have adequately addressed the reviewer's comments, making the manuscript significantly more impactful and compelling.

Point-by-Point Response to Reviewers' Comments

Overview

We thank the reviewers for their positive response and for recognizing the significance of our work. We are also grateful for the reviewers' rigorous feedback and suggestions on how to improve the manuscript. We have substantially revised the manuscript addressing all reviewers' comments to strengthen its publication in *Nature Communications*. Major revisions are: 1) We used dual-enSERT to examine fifteen new previously uncharacterized rare and common disease-linked variants. In doing so, we identified variants in *OTX2* and *MIR9-2* brain enhancers that reproducibly disrupt enhancer activity *in vivo*, implicating them in autism spectrum disorder and demonstrating the utility of dual-enSERT for studying uncharacterized non-coding variants; 2) We performed an additional single-cell RNA-seq experiment on F0 mice from the bicistronic dual-enSERT system and show no evidence for off-target effects; and 3) We reanalyzed this new single-cell RNA-seq data to unbiasedly identify genes potentially implicated in ectopic gene expression in the limb bud caused by candidate variants.

Please see our point-by-point response to the reviewers' comments below.

Reviewer 1

Reviewer 1, Overview: The manuscript by Hollingsworth and colleagues creates a new fluorescence reporter system to test for enhancer activity in mouse embryos. Given the exponential increase in whole-genome sequencing studies and the subsequent discoveries of thousands of non-coding variants, this reporter system is a valuable tool to test for function of those variants *in vivo*. The authors use 2 well defined enhancers, variants in which lead to polydactyly and associated with hypotonia, ataxia, and delayed development syndrome (HADDS) respectively. The authors also create a dual

fluorescent reporter with multiple insulators cloned in. This allows for the testing two enhancers simultaneously, hence facilitating the testing of risk and non-risk alleles from the same construct, creating in effect a heterozygous enhancer

We are grateful for the reviewer's positive feedback and recognition of the impact of our work.

Reviewer 1, Specific Comment 1: Though the authors perform single cell RNA seq study on the hindlimb bud of ZRSref-mCherry/ZRF404G>A-EGFP, the interpretation and analysis are very superficial (figure 3). The authors should ideally sort the mCherry+ cells, the GFP+ cells and double positive cells and redo the single cell assay. In absence of that the authors should at least subset out these cell (mCherry+, GFP+) populations from their current data and re-cluster them and then perform differential expression analysis. Their current analysis suggests there isn't much transcriptional differences between cells expression the wildtype and variant allele containing enhancers which I suspect is due to nature of the gene expression analysis.

We thank the reviewer for a suggestion to sort out cells and perform a deeper characterization of expressed genes in a subpopulation of fluorescent cells. The study's primary aim is to introduce a new *in vivo* method for testing enhancer variants and show its utility for testing new non-coding variants. In our opinion, an in-depth characterization of a subpopulation of ZRS-positive cells is outside of the scope of this study and is part of another manuscript focused on the mechanism of ZRS misregulation.

Nevertheless, we followed the reviewer's suggestion to subset and re-cluster the mCherry+ and eGFP+ cells as a more sensitive approach to detecting differentially expressed genes between these cell populations. We also performed an additional single-cell RNA-seq experiment (see response to **Specific Comment 6**) and included 6,000 additional cells in the analysis. Using this approach, we found a number of upregulated genes, such as *Asb4*, *Pax9*, and *Alx3*, that match the anterior-specific expression of variant-driven *eGFP*. We added these new results on **Pg 21 Ln 458** and as new **Fig. 5F-H** in the revised manuscript:

Results (Pg 21 Ln 458): To increase the sensitivity of the analysis we subset cells expressing *mCherry* and/or *eGFP* and re-clustered them (Fig. 5F). We found that cells clustered into those expressing both *mCherry* and *eGFP* (normal domain, Clusters 1 and 2) and those only expressing *eGFP* (ectopic domain, Cluster 3) (Fig. 5F, G). To identify genes specifically upregulated in eGFP-expressing cells, we performed differential analysis between Cluster 3 and all other clusters. We identified a number of anterior-biased genes, including *Asb4*, *Pax9*, and *Hpse2* (Fig. 5H), which match their expression patterns by in situ hybridization experiments of the limb bud. Taken together, these results implicate candidate pathways in ectopic hZRS activity and highlight how dual-enSERT enables capture of variant allele-labeled cells at cellular resolution.

Reviewer 1, Specific Comment 2: It is also unclear how many genes were significantly differentially expressed between these subclusters even in the current analysis? The heatmap appears to be made with cherrypicked set of genes. Beyond the transcription factors and A-P axis patterning genes what other genes are differentially expressed? This section on scRNA-seq analysis needs to be expanded and better explained.

In the revised manuscript, we now show a full heatmap that includes all ~200 differentially expressed genes between the three groups based on pairwise comparisons. We performed unsupervised hierarchical clustering on these genes to identify patterns of gene expression and found that, indeed, normal and ectopic cells largely overlap in their transcriptomes. We also expanded the results section to describe better the scRNA-seq analysis (**Pg 19 Ln 426**) and updated the heatmap in revised **Fig. 5E**:

Results (Pg 19 Ln 426): We next examined gene expression in cell subpopulations in which hZRS is normally active (*mCherry*⁺/*eGFP*⁺), ectopically active (*eGFP*⁺/*mCherry*⁻) and inactive (*mCherry*⁻/*eGFP*⁻). We performed unbiased differential gene expression analysis between these cell subpopulations to identify candidate genetic pathways linked to normal and ectopic *Shh* expression. With only thirty-six differentially expressed genes (FDR < 0.05, log₂FC > ±1.5), both normal and ectopic hZRS domains showed strong similarity in their transcriptional profiles, including an enrichment of known mesenchymal TFs such as *Msx1*, *Lhx2*, *Lhx9*, *Twist1* and others (Fig. 5E). This is consistent with the fact that many TFs specifying mesenchymal fate are expressed in the entire “progress zone” beneath apical ectodermal ridge (AER) which includes ectopic anterior and normal posterior domains of hZRS activity (Towers and Tickle 2009; Markman et al. 2023). By contrast, inactive cells expressed chondrocyte-specifying transcription factors like *Shox2* and *Sox9* (Fig. 5E) (Akiyama et al., 2002; Yu et al., 2007). Differential gene expression analysis between three clusters identified only a few genes such as *Asb4* that were specifically expressed in ectopic *eGFP*⁺/*mCherry*⁻ cells.

Reviewer 1, Specific Comment 3: It is also unclear in dual-enSERT-1 experiments how many copies of both the wild type and mutant enhancer are integrated in the H11 locus? Copy number would play a major role in number of cells expressing the fluorescent molecules and hence affect the single cell data.

We agree that the copy number is an important consideration for dual-enSERT-1. We performed qPCR to estimate the transgene copy number in each of the dual-enSERT-1 lines (The results were in the supplement of the original submission). Based on these results, we only compared enhancer alleles with the same copy numbers of integrated

transgenes. We have now highlighted this critical point in the manuscript's main text (see **Pg 5 Ln 113; Pg 7, Ln 152; Pg 10, Ln 225; Supplementary Figure 1A, B**).

Results (Pg 5 Ln 113): To visualize reference and variant hZRS enhancer activities simultaneously, we crossed these mouse lines to generate two-color dual-enSERT-1 embryos. Both lines had two copies of respective transgene integrated at the H11 locus, enabling direct and quantitative comparison between enhancer alleles in two-color dual-enSERT-1 embryos (See Fig. S1A, B and Methods).

Results (Pg 7 Ln 152): We generated a transgenic mouse line in which the human reference hs737 allele drives eGFP (hs737^{ref}-eGFP) and a second line in which an 830G>A variant allele, identified in a patient with autism, drives mCherry (hs737^{830G>A}-mCherry) (Fig. S1C). We then bred these mouse lines each containing single-copy integrated transgenes and examined reporter gene expression in E11.5 embryos (Fig. S1A, B).

Results (Pg 10 Ln 225): To visualize hZRS^{404G>A} and hZRS^{446T>A} variant allele activities simultaneously, we bred these mouse lines each containing two copies of respective transgene to generate two-color transgenic embryos (Fig. S1A, B).

Reviewer 1, Specific Comment 4: In figure 2 it is clear that the type of fluorescent molecule has an effect. In figure 2D (forelimb) 18% cells and 26% cells exclusively express 1 marker. It would be good for the authors to sort these single fluorescent labelled cells and do some sort of gene expression (even qPCR for selected markers including Shh and A-P axis marker) to see what if any are the transcriptional differences. This is an important piece of data to have for future studies when deciding on what fluorescent marker to pick.

We agree this is an important technical consideration and thank the reviewer for bringing it to our attention. To determine if these changes are due to inter-allelic variation or post-transcriptional changes, we performed their suggested experiment by sorting out anterior limb bud cells into four fluorescent populations: mCherry+/eGFP+, mCherry+ only, eGFP+ only, and mCherry-/eGFP- cells. We then performed RT-qPCR on cDNA libraries for the transgenes to determine mRNA levels for each cell population.

We found that eGFP+ cells accurately reflect *eGFP* mRNA expression (i.e., eGFP+ cells express significantly higher *eGFP* levels than eGFP- populations). We also observed that both mCherry+ cell populations express similar to *eGFP* expression levels of *mCherry*. However, we also detected *mCherry* expression in mCherry- cells. This may be explained by post-transcriptional differences, e.g., much slower maturation time and shorter half-life of the mCherry protein, compared to eGFP (Balleza et al, *Nat Methods* 2018; Khmelinskii et al. *Mol Cell Bio* 2016). Additionally, this incomplete fluorescent overlap may also be due to stochasticity from having the reporters on two separate alleles, as random allelic expression of autosomes is known to be pervasive, though untested at the H11 safe-harbor locus we use (Gimelbrant et al, *Science* 2007). Full qPCR results are shown below and now included as **Supplementary Figure 3C** and described in the Results on **Pg 8 Ln 190**:

Results (Pg 8 Ln 190): To test if this variability could be caused by differences between fluorophores, we performed *mCherry* and *eGFP* mRNA quantification in different populations of anterior limb bud cells. We dissected the anterior domains of hindlimbs from hZRS^{404G>A}-mCherry/hZRS^{404G>A}-eGFP embryos and sorted them into four cell populations: mCherry+/eGFP+, mCherry+/eGFP-, mCherry-/eGFP+, and mCherry-/eGFP-. We then performed qPCR to quantify *mCherry* and *eGFP* mRNA levels in each of these cell populations (Fig. S3C). Both eGFP-positive cell populations (mCherry+/eGFP+ and mCherry-/eGFP+) expressed more than 8-fold higher levels of *eGFP* than eGFP-negative cell populations (mCherry+/eGFP+ vs mCherry-/eGFP-, P = 0.008; mCherry+/eGFP+ vs mCherry+/eGFP-, P = 0.009) (Fig. S3C). This indicates that eGFP fluorescence accurately reflects *eGFP* expression.

Both mCherry-positive cell populations displayed comparable levels of eGFP and mCherry transcripts (P = NS). However, similar levels of *mCherry* transcripts were also observed in mCherry-negative cells (P = NS) (Fig. S3C). The incomplete fluorescent overlap on a single-cell level is likely due to post-transcriptional differences between fluorophores. For example, a significantly longer maturation time of mCherry in comparison to eGFP is consistent with eGFP+ cells expressing *mCherry* transcripts, but not

mature protein (Balleza et al. 2018; Khmelinskii et al. 2016). We mitigated the effect of this variability on our measurements of enhancer activity by quantifying fluorescence over the entire population of cells and using the heart as an endogenous control (Fig. 1D).

Reviewer 1, Specific Comment 5: The whole section “ZRS enhancer bypass three copies...” (line 269) should be moved to supplement and the authors should directly describe their dual-enSERT-2.0 study with synthetic insulators. They should direct the readers to the supplementary section about the single insulator study and its ineffectiveness.

We agree with the reviewer. We shortened the section about the failed 3xHS4 insulator in the main text (**Pg 11 Ln 251**) and instead expanded the Methods section (**Pg 28 Ln 609**). We also moved the entire figure describing these results to the supplement as **Fig. S4B, C**.

Reviewer 1, Specific Comment 6: The authors should ideally perform scRNA-seq studies on their single copy dual-enSERT-2.2 embryos for hZRS404G>A as a comparison to their studies with dual-enSERT-1. This will allow for the authors to judge any off target cellular effects from their new construct and give confidence of its efficacy and specificity.

Following the reviewer’s suggestion, we performed a scRNA-seq experiment on an age-matched hindlimb bud from a *dual-enSERT-2* embryo injected with B-hZRS^{ref}-*mCherry*/SI/hZRS^{404G>A}-*eGFP*-B to directly compare with our *dual-enSERT-1* results. We overlaid the datasets and found that almost every cell type is captured in the *dual-enSERT-2* version (see below). Most importantly, we observed the same cell populations express reference-driven *mCherry* (Distal Posterior) and variant-driven *eGFP* (Distal Middle, Distal Anterior, Proximal), suggesting little to no off-target effects in this transient *dual-enSERT-2* approach (**Fig. S8D**). We included these results on **Pg 18 Ln 402** and as new **Fig. S6A-D** in the revised manuscript:

Results (Pg 18 Ln 402): We performed two separate single-cell RNA-sequencing (scRNA-seq) experiments on dissected E11.5 hindlimb buds from F2 dual-enSERT-1 and F0 dual-enSERT-2 embryos, respectively (Figs. 1C, 3B, 5A and S8A). In both transgenic embryos, hZRS^{ref} allele drove *mCherry*, while hZRS^{404G>A} allele drove *eGFP*. To decrease reporter gene dropout (Kharchenko et al., 2014; Qiu, 2020), we adopted a nested PCR strategy to amplify *mCherry* and *eGFP* transcripts in our barcoded libraries (Methods; Fig. 5A) (Pollina et al., 2023). We processed scRNA-seq datasets generated from dual-enSERT-1 and dual-enSERT-2 hindlimb buds independently (Fig. S6A). Supervised clustering of dual-enSERT-1 hindlimb produced thirteen distinct cell types, including a large mesenchymal cluster defined by the specific expression of well-known marker genes (Fig. S6B) (Desanlis et al., 2020; Yokoyama et al., 2017). We recovered every cell type in dual-enSERT-2 hindlimb, except for proximal cells which we excluded during hindlimb dissection for dual-enSERT-2 (Fig. S6B). Having shown the reproducibility between dual-enSERT 1 and 2, we combined and integrated the two scRNA-seq datasets together to obtain approximately 21,000 cells and maximize statistical power for downstream analyses (Fig. 5B and S6C).

Reviewer 1, Specific Comment 7: Line 224, reference 41 is in a different reference style.

We updated the reference style for this reference.

Reviewer 2

Reviewer 2, Overview: Hollingsworth et al. reported improved versions of enSERT—an efficient CRISPR/Cas9-mediated site-specific transgenic mouse enhancer assay (Kvon et al. Cell 2020). To validate cell type-specific spatial enhancer activity, they developed dual-enSERT using two different reporter genes. To test the idea of dual-enSERT, they examined the impact of a known variant on ZRS enhancer activity and were able to analyze the enhancer activities with or without variants in the same mouse embryo using FACS sorting and single-cell analysis. To analyze two enhancer activities in mice, they also developed a bicistronic reporter assay (dual-enSERT-2.2) and confirmed the reproducibility of ZRS, hs737, and hs932 enhancer activity with or without variants.

I understand the concept of this study and recognize that analyzing spatial enhancer activity is crucial. This dual-enSERT system can precisely analyze variants' impact in mice *in vivo*. However, my impression of this work is that it is more incremental. I can see several technologically significant findings, such as the synthetic insulator and the addition of biotinylated nucleotides to obtain a single copy transgene. However, as the author mentioned, compared to other high-throughput assays like MPRA, dual-enSERT is very low throughput and it's difficult to see significant advancements compared to conventional enSERT or LacZ transgenic assays.

We thank the reviewer for their positive comments highlighting the strengths of the dual-enSERT assay and the critique. Given the exponential growth of WGS human genetics studies, expanding the toolkit of functional non-coding variant analysis remains a pivotal task for the field. While MPRA provides information about tens of thousands of variants, the MPRA results must be validated using *in vivo* assays, which are currently limited. Moreover, MPRA is not readily available in most developmental and disease contexts (e.g., limb development). Dual-enSERT thus complements current MPRA efforts by providing an orthogonal *in vivo* assay for prioritized variant validation, as was also highlighted by Reviewer #1.

In the revised manuscript, as was suggested by reviewer #2, we have significantly expanded the scope of the study by testing 15 previously uncharacterized variants from GWAS and WGS studies and identified variants that reproducibly affect enhancer activity *in vivo*, implicating them in autism spectrum disorder (see the response to Specific Comment 3).

Dual-enSERT provides a significant advance over the conventional enSERT or LacZ transgenic assay in three major ways: **1.** By placing both enhancer alleles on one transgene, we eliminate the impact of mosaicism, which reduces the reproducibility of enSERT and traditional LacZ assays. We show that only a few dual-enSERT-2 embryos are needed to reproducibly detect a change in enhancer activity (e.g., **Fig. 4**). In contrast, with enSERT, two separate transgenic constructs need to be injected. Often tens of embryos need to be analyzed for each line to eliminate the effects of mosaicism (see, e.g., a recent Nat Genet study from E. Engle lab: PMID: 37386251). **2.** Dual-enSERT enables quantitative comparison of enhancer allele activities, which is not possible with enSERT and traditional LacZ assays. This is especially critical for common variants whose effects on enhancer activity are expected to be more subtle (see, for example, our new result for the autism-linked STR polymorphism addressing Specific Comment 3). **3.** Dual-enSERT enables access to the cells in which variants cause an enhancer to be inactive (loss-of-function) or ectopically active (gain-of-function) for downstream applications, such as expression profiling and epigenomic analysis. In the revised manuscript, we better highlight the novelty of our new method.

We hope the new transgenic, scRNA-seq, and qPCR experiments and text revisions sufficiently address the reviewer's comments.

Reviewer 2, Specific Comment 1: Given its broad readership, including non-mouse geneticists, I would like to see more data demonstrating a pathogenic mechanism with this system for publication in Nature Communications. Although the authors tested three variants associated with polydactyly, autism, and craniofacial malformations, the conventional LacZ transgenic assay has already been performed in previous work as cited in the manuscript, and it's difficult to see significant novel findings from this paper.

We agree with the reviewer that understanding the pathogenic mechanisms of enhancer variants is a fundamental and pressing question. To address this point, we performed an additional scRNA-seq experiment from an *dual-enSERT-2* F0 embryo with the 404G>A variant and performed additional data analysis on a significantly expanded number of cells (21,000 cells in comparison to 15,000 cells in the original submission). This analysis revealed that both normal and ectopic domains of ZRS enhancer activity showed strong similarity in their transcriptional profiles, including enrichment of known mesenchymal TFs such as *Msx1*, *Lhx2*, *Lhx9*, *Twist1* and others (see heatmap below). This is consistent with the fact that many TFs specifying mesenchymal fate are expressed in the entire “progress zone” beneath apical ectodermal ridge (AER) which includes ectopic anterior and normal posterior domains of ZRS activity providing potential explanation of enhancer misactivation upon point mutations. We added the new results on **Pg 18 Ln 402** and as new **Fig. 5E** in the revised manuscript:

Results (Pg 18 Ln 402): We performed two separate single-cell RNA-sequencing (scRNA-seq) experiments on dissected E11.5 hindlimb buds from F2 dual-enSERT-1 and F0 dual-enSERT-2 embryos, respectively (Figs. 1C, 3B, 5A and S8A). In both transgenic embryos, hZRS^{ref} allele drove *mCherry*, while hZRS^{404G>A} allele drove *eGFP*. To decrease reporter gene dropout (Kharchenko et al., 2014; Qiu, 2020), we adopted a nested PCR strategy to amplify *mCherry* and *eGFP* transcripts in our barcoded libraries

(Methods; Fig. 5A) (Pollina et al., 2023). We processed scRNA-seq datasets generated from dual-enSERT-1 and dual-enSERT-2 hindlimb buds independently (Fig. S6A). Supervised clustering of dual-enSERT-1 hindlimb produced thirteen distinct cell types, including a large mesenchymal cluster defined by the specific expression of well-known marker genes (Fig. S6B) (Desanlis et al., 2020; Yokoyama et al., 2017). We recovered every cell type in dual-enSERT-2 hindlimb, except for proximal

cells which we excluded during hindlimb dissection for dual-enSERT-2 (Fig. S6B). Having shown the reproducibility between dual-enSERT 1 and 2, we combined and integrated the two scRNA-seq datasets together to obtain approximately 21,000 cells and maximize statistical power for downstream analyses (Fig. 5B and S6C).

We next examined gene expression in cell subpopulations in which hZRS is normally active (*mCherry*⁺/*eGFP*⁺), ectopically active (*eGFP*⁺/*mCherry*⁻) and inactive (*mCherry*⁻/*eGFP*⁻). We performed unbiased differential gene expression analysis between these cell subpopulations to identify candidate genetic pathways linked to normal and ectopic *Shh* expression. With only thirty-six differentially expressed genes (FDR < 0.05, log₂FC > ±1.5), both normal and ectopic hZRS domains showed strong similarity in their transcriptional profiles, including an enrichment of known mesenchymal TFs such as *Msx1*, *Lhx2*, *Lhx9*, *Twist1* and others (Fig. 5E). This is consistent with the fact that many TFs specifying mesenchymal fate are expressed in the entire “progress zone” beneath apical ectodermal ridge (AER) which includes ectopic anterior and normal posterior domains of hZRS activity (Towers and Tickle 2009; Markman et al. 2023). By contrast, inactive cells expressed chondrocyte-specifying transcription factors like *Shox2* and *Sox9* (Fig. 5E) (Akiyama et al., 2002; Yu et al., 2007). Differential gene expression analysis between three clusters identified only a few genes such as *Asb4* that were specifically expressed in ectopic *eGFP*⁺/*mCherry*⁻ cells.

We also subset and re-clustered the *mCherry*⁺ and *eGFP*⁺ cells as a more sensitive approach to detecting differentially expressed genes between these cell populations.

Using this subset approach, we found a number of upregulated genes, such as *Asb4*, *Pax9*, and *Alx3*, that match the anterior-specific expression of variant-driven *eGFP*. We added the new results on **Pg 21 Ln 458** and as new **Fig. 5F-H** in the revised manuscript:

Results (Pg 21 Ln 458): To increase the sensitivity of the analysis we

subset cells expressing *mCherry* and/or *eGFP* and re-clustered them (Fig. 5F). We found that cells clustered into those expressing both *mCherry* and *eGFP* (normal domain, Clusters 1 and 2) and those only expressing *eGFP* (ectopic domain, Cluster 3) (Fig. 5F, G). To identify genes specifically upregulated in *eGFP*-expressing cells, we performed differential analysis between Cluster 3 and all other clusters. We identified a number of anterior-biased genes, including *Asb4*, *Pax9*, and *Hpse2* (Fig. 5H), which match their expression patterns by in situ hybridization experiments of the limb bud. Taken together, these results implicate candidate pathways in ectopic hZRS activity and highlight how dual-enSERT enables capture of variant allele-labeled cells at cellular resolution.

We would like to highlight that the primary strength and focus of this manuscript is in its technical advance: accelerating the pace of *in vivo* testing for non-coding variants. To address the reviewer's criticism and reinforce our new tool's utility, we have interrogated fifteen untested rare human and common variants linked to neurodevelopmental disorders, as suggested by this reviewer in Specific Comment 3 (see Specific Comment 3 for related data). We have also highlighted it in the abstract and the discussion and toned down parts that claimed the identification of pathogenic mechanisms.

Reviewer 2, Specific Comment 2: I really appreciate the idea of using this system to understand the mechanistic difference between 'gain of activator' versus 'loss of repressor' as shown in Figure 2. It's interesting that two different mutations cause the same polydactyly condition. The authors may want to conduct single-cell analysis for the 446T>A (gain of activator) variant with the same system as in Figure 3 and compare the cell subpopulations. As the author mentions on page 7, line 168, I am curious whether these independent mutations result in ectopic gene expression in the same or different cell populations of the limb bud.

We agree that identifying whether these are the same or different populations is an important question. Our data in **Figure 2B** and the more sensitive FACS-based analysis of cells in **Figure 2C, D** convincingly show that the same cell populations are affected by the two different variants. The percentage of cells in which both enhancer variants are ectopically active is statistically indistinguishable from a control experiment in which the same enhancer variant drives both *mCherry* and *eGFP* (**Fig. 2A, D**).

Because of gene dropout and the low coverage of scRNA-seq, FACS-based experiments performed in **Fig. 2D** are more sensitive to capturing populations expressing specific fluorescent reporters. Thus, a costly new scRNA-seq experiment for 446T>A variant will not reveal significant new insights.

However, we performed an additional scRNA-seq experiment for the 404G>A variant, which added 6,000 more cells and enabled a more robust analysis (see response to the previous comment).

Reviewer 2, Specific Comment 3: As they have developed a rapid enhancer assay system, I would like them to explore uncharacterized variants identified from GWAS as an example. This might be out of scope, but it would certainly provide more valuable data and demonstrate the system's effective application.

We thank the reviewer for this valuable suggestion and agree this would effectively demonstrate its widespread utility to the field. We have tested fifteen additional previously uncharacterized human non-coding variants linked to neurodevelopmental disorders and have identified both rare and common variants that reproducibly disrupt enhancer activity (both causing loss- and gain-of-activity effects, respectively). Full details on the variants tested and these new data are discussed in the new result section (**Pg 15 Ln 345**) and included as **Table 1, Figure 4**, and **Supplementary Figure 5**, of the revised manuscript, all of which are appended below for reviewer convenience:

Results (New Section, Pg 15 Ln 345): Scaled assessment of previously uncharacterised non-coding variants with dual-enSERT-2

We next sought to use dual-enSERT-2 to functionally screen previously uncharacterised human non-coding variants linked to congenital disease. We compiled a list of thousands of candidate pathogenic rare and common non-coding variants from patients with neurodevelopmental disorders identified by GWAS and whole genome sequencing studies (Iossifov et al. 2014; Turner et al. 2017; Niemi et al. 2018; Short et al. 2018; Grove et al. 2019; Shin et al. 2023). We intersected this list with previously validated in vivo human and mouse enhancers active at embryonic day E11.5. We further narrowed the list by focusing on enhancers affected by multiple independent non-coding variants or enhancers with a known NDD-linked target gene based on capture Hi-C data, reasoning these regions the least likely to be random mutations (Fig. 4A, Table 1 and Methods) (Chen et al. 2024). From this prioritization we chose fourteen rare SNVs and one common indel distributed between seven unique enhancers active in the forebrain, midbrain, hindbrain, or neural tube.

To efficiently test these prioritized variants we generated compound variant alleles for each of the enhancers, in some instances up to three independent variants per enhancer (Fig. 4A). We placed these compound variant alleles upstream of eGFP and the corresponding reference enhancer alleles upstream of mCherry and compared them using dual-enSERT-2. We observed loss of enhancer activity in one enhancer (hs268), gain of enhancer activity for one enhancer (hs1791), no detectable changes for four enhancers (Fig. 4 and Fig. S5A-H), and one enhancer (hMM1518) was inactive in our assay possibly due to interspecies sequence divergence (Fig. S5I). For example, introduction of two rare, autism-linked 435G>T and 700C>T variants in the hs268 enhancer of *MIR9-2* resulted in substantial loss of enhancer activity in the brain and neural tube (5/5 embryos; 3.6-fold in the

forebrain, $P = 0.001$; 3.2-fold in the midbrain, $P = 0.0093$; 3.3-fold in the hindbrain, $P = 0.0065$; 3.4-fold in the neural tube, $P = 0.0056$; Fig. 4B, C) . 435G>T and 700C>T variants disrupt evolutionary conserved putative binding sites for neuronally-expressed TFs TBX/TBR2 and BCL11A, respectively (Fig. 4D). Conversely, a variant allele of the hs1791 midbrain enhancer of *OTX2* containing a common STR polymorphism linked to autism (rs1880784044) resulted in an almost two-fold increase in midbrain enhancer activity (4/4 embryos; $P = 0.011$; Fig. 4E-G) (Grove et al. 2019). These results indicate that dual-enSERT-2 can be used for rapid functional screening of non-coding variants linked to congenital disorders.

Table 1 (New). Summary of *dual-enSERT-2* results for previously uncharacterised human enhancer variants.

Variant Location (hg38)	dbSNP ID	Associated Disorder	Enhancer (Putative Target Gene(s))	Normal Enhancer Activity	Reproducible Changes in Enhancer Activity	Reference
chr1:213425381T>C, chr1:213425533A>C	- -	Autism spectrum disorder	hs204 (PTPN14)	Forebrain	No change	Shin et al, 2024
chr5:88396771G>T, chr5:88397035C>T	rs576375513, -	Autism spectrum disorder	hs268 (MIR9-2)	Forebrain, Midbrain, Hindbrain, and Neural Tube	Loss of activity in Forebrain, Midbrain, Hindbrain and Neural Tube	Shin et al, 2024
chr19:30843449G>C, chr19:30843509G>A	- -	Neurodevelopmental disorder	hs430 (-)	Midbrain	No change	Short et al, 2018
chr3:180462583T>C, chr3:180462704A>G	rs189450851, -	Neurodevelopmental disorder	hs655 (RNF220 , ERI3 , DMAP1)	Hindbrain	No change	Short et al, 2018
chr3:147847042T>C, chr3:147847133T>C, chr3:147847216T>C	-, -, rs1166255572	Autism spectrum disorder	hs1573 (-)	Forebrain, Hindbrain	No change	Shin et al, 2024
chr14:57007962CGA AGA>C	rs1880784044	Autism spectrum disorder	hs1791 (OTX2)	Midbrain	Gain of activity in Midbrain	Grove et al, 2019
chr15:66774248C>T, chr15:66774318C>T, chr15:66774321C>T	- -, rs546228412	Autism spectrum disorder	hMM1518 (SMAD6 , SCARLETLTR)	Not active	No change	Zhuo et al, 2019

Reviewer 2, Specific Comment 4: Page 9, Line 206: The values for '50% (in forelimbs) to 56% (in hindlimbs)' are swapped between the forelimbs and hindlimbs.

We thank the reviewer for pointing out this and have corrected this mistake.

Reviewer 2, Specific Comment 5: When citing Fig. S3, please indicate parts A-G; it's hard to follow. Currently, there is no indication of A-G in the main text."

We apologize for this ambiguity. We have explicitly called out the panels when discussing the related data (now Supplementary Figure 4) in the Results.